# Active DNA unwinding and transport by a membrane-adapted helicase nanopore

Ke Sun[1,8], Changjian Zhao[1,8], Xiaojun Zeng[1,8], Yuejia Chen[1,8], Xin Jiang[1,8], Xianting Ding[2], Lu Gou[3], Haiyang Xie[2], Xinqiong Li[1], Xialin Zhang[1], Sheng Lin[1], Linqin Dou[1], Long Wei[1], Haofu Niu[1], Ming Zhang[1], Ruocen Tian[1], Erica Sawyer[1], Qingyue Yuan[1], Yuqin Huang[4], Piaopiao Chen[1], Chengjian Zhao[1], Cuisong Zhou[4], Binwu Ying[1], Bingyang Shi[5], Xiawei Wei[1], Ruotian Jiang[6], Lei Zhang [3]*, Guangwen Lu [7]* & Jia Geng[1]*

Nanoscale transport through nanopores and live-cell membranes plays a vital role in both key biological processes as well as biosensing and DNA sequencing. Active translocation of DNA through these nanopores usually needs enzyme assistance. Here we present a nanopore derived from truncated helicase E1 of bovine papillomavirus (BPV) with a lumen diameter of c.a. 1.3 nm. Cryogenic electron microscopy (cryo-EM) imaging and single channel recording confirm its insertion into planar lipid bilayer (BLM). The helicase nanopore in BLM allows the passive single-stranded DNA (ssDNA) transport and retains the helicase activity in vitro. Furthermore, we incorporate this helicase nanopore into the live cell membrane of HEK293T cells, and monitor the ssDNA delivery into the cell real-time at single molecule level. This type of nanopore is expected to provide an interesting tool to study the biophysics of biomotors in vitro, with potential applications in biosensing, drug delivery and real-time single cell analysis.

[1] Department of Laboratory Medicine, State Key Laboratory of Biotherapy, National Clinical Research Center for Geriatrics, West China Hospital, Sichuan University and Collaborative Innovation Center for Biotherapy, Chengdu 610041, China. [2] Institute for Personalized Medicine, State Key Laboratory of Oncogenes and Related Genes, School of Biomedical Engineering, Shanghai Jiao Tong University, Shanghai 200030, China. [3] MOE Key Laboratory for Nonequilibrium Synthesis and Modulation of Condensed Matter, School of Science, Xi'an Jiaotong University, Xi'an 710049, China. [4] College of Chemistry, Sichuan University, Chengdu 610041, China. [5] Henan-Macquarie Uni Joint Center for Biomedical Innovation, School of Life Sciences, Henan University, Kaifeng, Henan 475004, China. [6] Laboratory of Anesthesia and Critical Care Medicine, Department of Anesthesiology, West China Hospital, Sichuan University, Chengdu, Sichuan 610000, China. [7] West China Hospital Emergency Department (WCHED), State Key Laboratory of Biotherapy, West China Hospital, Sichuan University, Chengdu, Sichuan 610041, China. [8] These authors contributed equally: Ke Sun, Changjian Zhao, Xiaojun Zeng, Yuejia Chen, Xin Jiang. *email: zhangleio@mail.xjtu.edu.cn; lugw@scu.edu.cn; geng.jia@scu.edu.cn

Molecular transport through nanoscale channels including membrane channels, biological nanopores, and artificial nanopores are of great importance in several key biological processes[1] and engineering applications[2]. For single-molecule sensing[3,4], DNA[5,6], and RNA[7] sequencing, various types of nanopores have been constructed, including protein nanopore from bacteria[8–12], viral portal protein[13], inorganic materials[14–17], and DNA origami[18,19]. However, translocation of DNA through these nanopores itself is usually driven by an external force such as voltage, and the nanopore functions are primarily studied in vitro. Enzyme-assisted unwinding of double-stranded DNA (dsDNA) through nanopore was realized[20,21] and designed for DNA sequencing applications ingeniously[22].

Helicases play a vital role for all living organisms to unpackage their genes fueled by ATP hydrolysis[23,24]. The BPV E1 protein is a hexamer helicase from bovine papillomavirus (BPV). It has a central pore with lumen diameter of 1.1–1.5 nm[25]. The wild-type BPV E1 contains N-terminal domain (residues 1–141), DNA binding domain (DBD) (residues 142–307), helicase domain (residues 308–577), and C-terminal domain (residues 578–605)[26,27]. BPV E1 helicase plays an important role in the virus genome replication, during the unwinding process, the helicase moves along the one strand of the dsDNA in the direction of 3′–5′ powered by ATP hydrolysis. BPV E1 helicase with DBD can bind to the specific sequence of the viral genome DNA (origin of DNA replication) with the assistance of E2 transcription factor and start the unwinding of genomic DNA in vivo. While in vitro, the truncated BPV E1 (without DBD) has no sequence specificity and needs single strand arm of dsDNA for the binding and unwinding[28]. The 28-aa peptide C-terminal is important for the E1 complex formation[27].

Single-molecule studies provide a powerful tool revealing the detailed dynamics of biomotors, including DNA unwinding process by helicases, such as optical tweezers[29], fluorescence resonance energy transfer (FRET) microscopy[30,31], nanopore[32], and DNA curtains[33]. The real-time study on kinetics of enzyme at high spatial resolution and time resolution have been reported on motor proteins, including helicase[21,34], polymerase[20], and other proteins such as exonuclease I[35], and single-base resolution at the single-molecule level could be realized[20].

Inspired by the structural similarity of helicase E1 from BPV and other hexameric transmembrane channels, here we report that BPV E1 can be truncated into a helicase nanopore and inserted into a planar lipid bilayer. Its membrane compatibility has been studied with liposome vesicles using cryo-EM, reconstitution into planar lipid bilayer by single channel recording. In high-salt conditions ssDNA could be transported based on an external applied voltage, while in physiological conditions dsDNA containing a single-stranded arm could be unwound and transported through the helicase nanopore in a way that is comparable with helicase activity in vitro. And meanwhile, this helicase nanopore is introduced to cell electrophysiology realizing real-time ssDNA transfection into a HEK293T cell as well as macromolecule extraction from inside the cell. The helicase nanopore enables potential applications in single-molecule enzyme kinetics, biosensing, and live cell analysis in situ.

## Results

### Construction of the helicase nanopore and membrane fusion.
The hexameric BPV E1 possesses a nanoscale lumen and shares structural similarities with commonly used protein nanopores such as MspA and α-hemolysin, except that there is no transmembrane domain in BPV E1. To evaluate the possibility of reconstructing a membrane-adapted helicase nanopore based on BPV E1, the electrostatic surface was shown in Fig. 1a, b and the

hydrophobic residues were shown in green in Supplementary Fig. 1. It could be observed that more hydrophobic residues spread over the middle of the facing the N-terminal (panel b left), which may facilitate the adhesion and insertion of the protein into the lipid bilayer membrane. In order to increase its solubility in aqueous solution, a glutathione S-transferase (GST)-tag was added to the N-terminal. In this study, two kinds of truncated BPV E1 protein containing 306–577 fragment (BPV E1 helicase domain) and 306–605 fragment (BPV E1 helicase domain and C-terminal tail) were reconstructed with N-terminal GST fusion protein into pGEX-6P-1 plasmid and expressed in *Escherichia coli* (*E. coli*), respectively (Supplementary Table 1, Fig. 2a, b). Both expressed proteins were purified by size exclusion chromatography (Supplementary Fig. 2c, d) and then identified by sodium dodecyl sulfate polyacrylamide gel electrophoresis (SDS-PAGE) (Fig. 1c). Compared with $E1_{306-577}$ (with GST-tag) and $E1_{306-605}$ (with GST-tag cleaved) which tended to precipitate, $E1_{306-605}$ with GST-tag was more stable. Hence, we chose truncated $E1_{306-605}$ protein with GST-tag as helicase nanopore in the following studies unless otherwise specified.

To verify whether the helicase nanopore could be inserted into the lipid bilayer membrane or not (as shown in Fig. 1d), we prepared proteoliposome vesicles containing the helicase nanopore for cryo-EM characterization (Fig. 1e, Supplementary Fig. 3). Synthetic lipid diphytanoyl phosphatidylcholine (DPhPC) was used in this study for vesicle and planar lipid bilayer preparation unless otherwise specified. High resolution images of 67 helicase nanopores from 45 independent images were collected and analyzed, and red solid lines and blue dashed lines highlighting the bilayer membrane and helicase nanopore, respectively were provided in the insets. Compared with the control sample of blank liposome (Fig. 1e left), those proteoliposome vesicle images provided direct evidences of helicase nanopore insertion into lipid bilayer membrane (Fig. 1e right). Further detailed analysis was performed on the 67 membrane-embedded helicase nanopores. The perpendicular orientation of the helicase nanopore to the membrane plane was strongly preferred, with majority of the helicase nanopore tilting close to vertical from the membrane (Supplementary Fig. 4a). As shown in Supplementary Fig. 4b, the numbers of inserted/attached helicase nanopore from outside the vesicles and from inside the vesicles were close. It can be concluded that about half of the helicase nanopores were attached on the bilayer surface (similar to the truncated α-hemolysin pores[36]), and another half were inserted into the bilayer with varying depths (Supplementary Fig. 4c). Statistical analysis revealed a relatively wide distribution of inserted lengths of the helicase nanopores into vesicle membrane ($4.7 \pm 0.6$ nm, the error bars indicated standard deviation in this work unless otherwise specified, Supplementary Fig. 4d), and 83.5% of the insertion were N-terminal incorporation (Supplementary Fig. 4e). These observations and analysis provided interesting information of the interaction of helicase nanopore with bilayer membranes in proteoliposome vesicles.

### ssDNA translocation through the helicase nanopore in BLM.
The conductivity of the helicase nanopore in the lipid bilayer was performed using a single channel conductance assay setup[11] (Fig. 2a). Solution containing the helicase nanopore (final concentration of 2–10 ng/mL) was added to the buffer solution in the *cis*-side of the BLM chambers, which was connected to the ground electrode. When a single helicase nanopore was inserted into the membrane, an instantaneous current increase can be observed as shown in a current trace under either a positive or negative bias voltage (Fig. 2b, 0.5 M KCl, 5 mM HEPES, pH 7.5). Occasionally, larger current increase was observed as shown in a

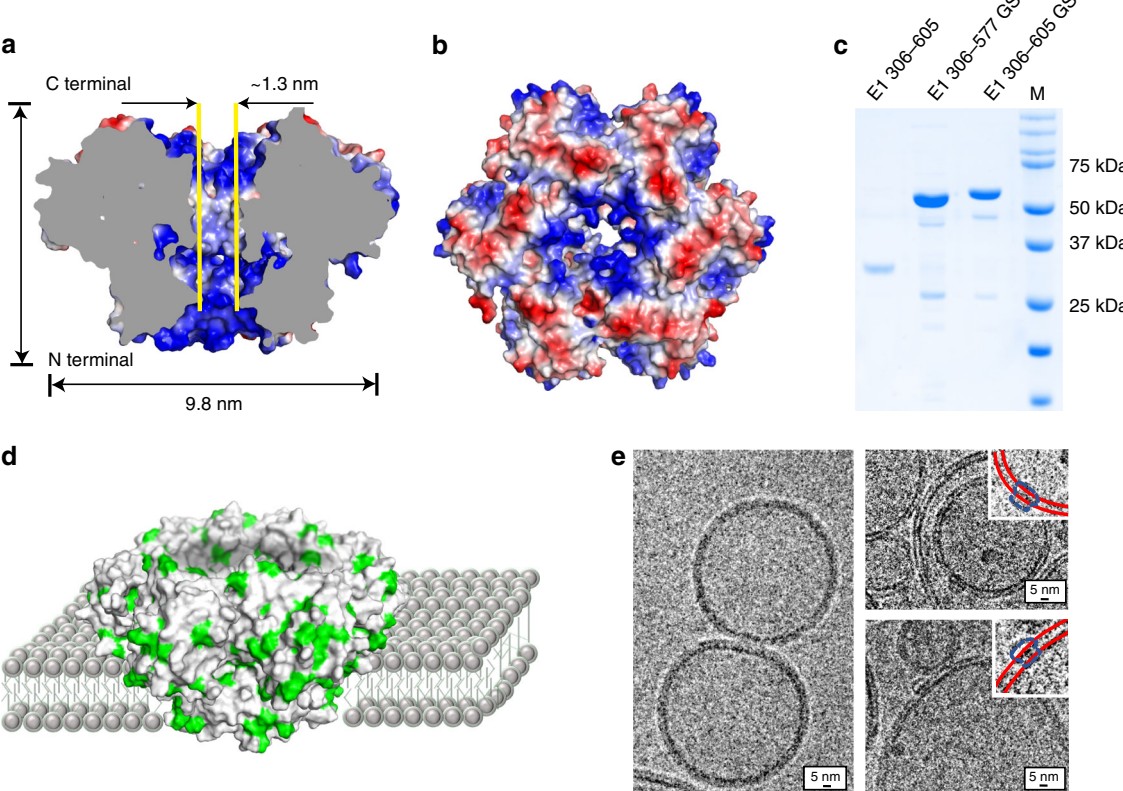

**Fig. 1** The structure of BPV E1 helicase nanopore and its incorporation into liposomes. **a**, **b** Top and side view of the helicase nanopore showing a central lumen with diameter of c.a. 1.3 nm[28] (PDB ID: 5a9k). Coloring scheme of red for negative, white for neutral, and blue for positive electrostatic potential was used. **c** Coomassie-blue stained SDS-gel showing different constructs of helicase nanopore from truncated BPV E1. **d** A schematic drawing illustrating the insertion of a helicase nanopore into a lipid bilayer, with protein hydrophobic residues colored in green. According to the analysis of membrane insertion results, the helicase nanopore can be inserted into lipid bilayer membrane in either orientation. N-terminal insertion case was shown in the illustration. **e** Cryo-EM image of DPhPC liposome vesicles containing helicase nanopores. Left: Control sample of liposome vesicle only; Right: C-terminal (top right) and N-terminal (down right) insertion into the lipid bilayer of the liposome vesicle (n = 67 from 46 trials). Insets highlighting bilayer membrane (red solid lines) and helicase nanopore (blue dashed lines) were provided in a magnified view. Source data used to generate panels **c** and **e** in this figure can be found in the Source Data file

current recording of 760 s with three pores inserted simultaneously in Supplementary Fig. 5 (bias voltage of +70 mV in the buffer solution of 1 M KCl, 5 mM HEPES, pH 7.5), which could be attributed to the simultaneous insertion of helicase nanopores. The Gaussian fit to the conductance histogram for Fig. 2c gave the peak value of 1.34 ± 0.03 nS in the buffer solution of 1 M KCl, 5 mM HEPES, pH 7.5 under a bias voltage of +70 mV, and was 0.23 ± 0.01 nS in phosphate buffered saline (PBS) (n = 34 from 19 trials). The conductance measurements were also performed under a ramping voltage. The slope from the fitted curve was *c.a.* 1.3 nS for a single pore in buffer solution of 1 M KCl, 5 mM HEPES, pH 7.5 (Fig. 2d).

We also compared the measured conductance with the value calculated using the equation from Hall[37]. The equation was used on the alpha-hederin[38] nanopore sharing similar lumen characteristics with the helicase nanopore. The variation of the conductance as a function of the diameter can be described as below

$$G_{\text{nanopore}} = \sigma_{\text{KCl}} \left( \frac{4h}{\pi d^2} + \frac{1}{d} \right)^{-1}, \quad (1)$$

where $\sigma_{\text{KCl}}$ (11.1 S/m) is the molar conductivity of the electrolyte (1 M KCl), $h$ is the effective thickness of the nanopore, and $d$ is the diameter of the nanopore. Effective mean thicknesses of 6.7 nm and the diameter range from 1.1 to 1.5 nm for helicase

nanopore were obtained from its crystal structure[25,28,39]. The calculated conductance for the helicase nanopore in 1 M KCl solution (electrolyte conductivity 11.1 S/m) was 1.39–2.49 nS. This is comparable to the measured conductance value (1.34 ± 0.03 nS). The pore conductance was compared to that of α-hemolysin under the same condition.

The membrane insertion efficiency of the helicase nanopores with and without the GST-tag into planar lipid bilayer were tested, where the insertion efficiency was defined as the ratio of number of trials with helicase nanopore inserted into lipid bilayer to the total number of trials. From the statistics of 672 independent trials of single channel recording experiments, the efficiency of insertion was *c.a.* 44.8% (Supplementary Fig. 6). The results showed that the helicase nanopore with GST-tag (253 out of 494 trials, 51.2%) was more membrane-compatible than that without GST-tag (48 out of 178 trials, 27.0%). Supplementary Fig. 7 showed the scatter diagrams of the time consumption of the helicase nanopore insertion and the duration time of helicase nanopore embedded in BLM. We also investigated the helicase nanopore insertion using *E. coli* total lipid extract besides DPhPC. Under similar experiment conditions, two types of lipid component showed similar insertion efficiency and membrane stability.

To verify the capability of ssDNA translocation through the helicase nanopore, we added ssDNA (sequence A of 48 nt, Supplementary Table 2) into the buffer solution to *cis*-chamber

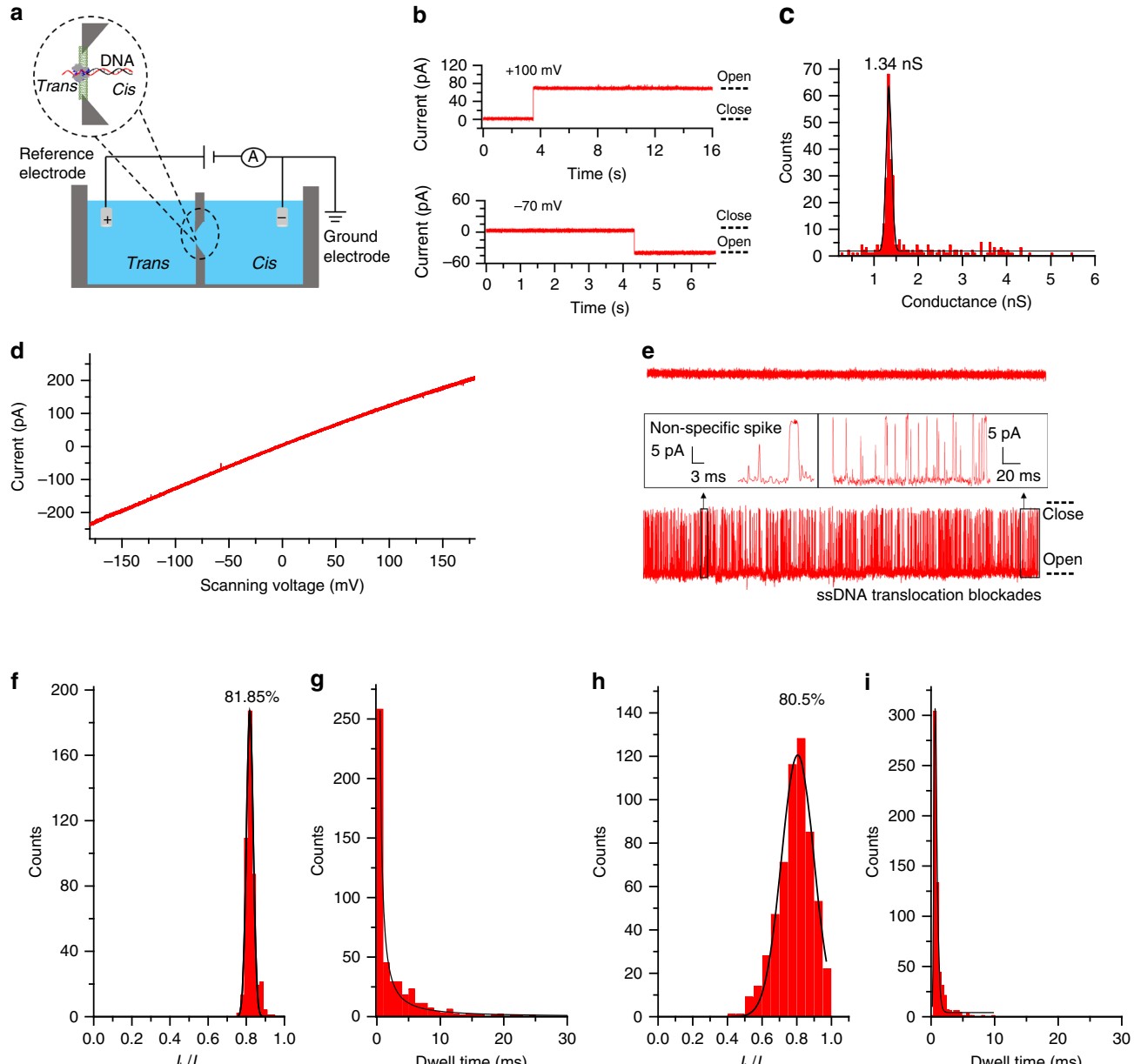

**Fig. 2** Conductance assay and ssDNA translocation through the helicase nanopore in planar lipid bilayer. **a** The illustration of bilayer setup for single channel recording. **b** Current traces recording the single incorporation of the helicase nanopore into lipid bilayer. The buffer conditions were 0.5 M KCl, 5 mM HEPES, pH 7.5 and the bias voltage was +100 mV (top) and −70 mV (bottom). **c** Histogram of the conductance of the helicase nanopore in the lipid bilayer. The buffer condition was 1 M KCl, 5 mM HEPES, pH 7.5 and the bias voltage was +70 mV ($n = 320$ from 15 independent trials). **d** I–V curve of helicase nanopore in a ramp voltage ranging from −180 to +180 mV (Buffer conditions: 1 M KCl, 5 mM HEPES, pH 7.5). **e** Current traces of helicase nanopore in BLM without (top) and with 100 nM ssDNA (Sequence A of 48 nt, from 5 independent trials). Buffer condition: 0.5 M KCl, 5 mM HEPES, pH 7.5 under −50 mV. **f, g** Histograms showing the percentage of current blockages and dwell times caused by linear ssDNA under +50 mV (Buffer condition: 0.5 mM KCl, 5 mM HEPES, pH 7.5, $n = 453$ from 5 trials). **h, i** Histograms showing the percentage of current blockages and dwell times caused by linear ssDNA under +100 mV (Buffer condition: 0.5 mM KCl, 5 mM HEPES, pH 7.5, $n = 580$ from 5 trials). Source data generating panel **b** to **i** are provided as a Source Data file

under a bias voltage of −50 mV. It was found that the addition of 100 nM ssDNA (sequence A of 48 nt, Supplementary Table 2) could induce a large number of current blockages after a single pore insertion (Fig. 2e). The residual current of the blockage events could be obtained by $I_r = I_o − I_b$, where $I_r$ represented the residual current when the pore was blocked, $I_b$ was the blocked current, and $I_o$ was the open pore current. Analysis of the current blockage events indicated that the ssDNA caused both distinct blockage events, and nonspecific spikes due to occasional

background noise or ssDNA random collision with the helicase nanopore. Under +50 mV in 0.5 M KCl buffer solution, Gaussian fitting of these current blockage events gave the peak value of ($I_b/I_o$) 81.850 ± 0.001% (Fig. 2f) and a mean dwell time of 1.1 ± 0.2 ms (Fig. 2g). After changing the bias voltage to +100 mV, the events frequency increased with a mean dwell time of 0.64 ± 0.02 ms (Fig. 2h) and a peak value of 80.5 ± 0.5% (Fig. 2i). This is comparable to the report that oligo-ssDNA immobilized inside the lumen of α-hemolysin induced current blockages

ranging from 50% to 90% due to the different compositions of nucleotides[40].

In order to verify that the resistance pulse-like blockage events appearing in the current traces were indeed caused by ssDNA, a nuclease degradation experiment was implemented. Interestingly, when DNase I was added into the *cis*-chamber (0.25 μg/mL) following the addition of ssDNA (Sequence B of 80 nt, 200 nM), the large number of transient blockages decreased after 20 min, and hardly signals could be observed after 50 min (0.5 M KCl, 5 mM HEPES, pH 7.5, under +50 mV, Supplementary Fig. 8). This result confirmed that the blockage signals was induced by ssDNA. To rule out the impact of nonspecific event signals brought by transient binding and dissociation, we carried out additional concentration-dependent ssDNA translocation experiments (Supplementary Fig. 9, 0.5 M KCl, 5 mM HEPES, pH 7.5). As shown in panel a and b, with ssDNA concentration increased from 25 pM to 25 nM, the translocation events frequency showed linear increase. Moreover, with the bias voltage increased from +50 to +100 mV, the dwell time of blockage events and interevent gap time reduced significantly while the events frequency increased (panels c–e). The frequency of DNA translocation through protein nanopore could be increased by augmenting the internal positive charge of the nanopore[41], and it was interesting to found that the frequency of ssDNA transloca-tion events through the helicase nanopore was higher compared with α-hemolysin even with internal positive charge augment under similar condition[41]. The reason for faster DNA capture rates could be attributed to the nature of the virus-derived helicase motor to process and unwind dsDNA in host cell during infection, in contrast to the bacteria derived pore-forming toxin nanopores. These data demonstrated that the blockage signals were attributed to ssDNA translocation rather than random collisions.

The above results confirmed that the helicase nanopore could be inserted into the lipid bilayer membrane similar to other reported protein nanopores with the ability of passive transloca-tion of ssDNA driven by electric field force.

**Unwinding of dsDNA by the helicase nanopore in vitro**. It is reported that BPV E1 belongs to the super family 3 (SF3) class of helicases, and DNA enter the BPV E1 helicase in the direction of 3′–5′[28]. We measured the unwinding activity of the helicase nanopores by in vitro helicase assay[42]. A dsDNA substrate was designed, with quencher BHQ2 and fluorophore Cy3 at each end of one of its strands (sequence C of 22 bp, structure shown in Supplementary Fig. 10). Totally, 5 nM DNA substrate was incu-bated with 50 nM helicase nanopore in the reaction buffer at 37 ° C for 30 min. During the first 10 min, the baseline fluorescence was measured as a background signal. After the addition of 2 mM ATP, the fluorescence gradually decreased due to fluorescence quenching (Fig. 3a). This result can be explained that the helicase motor first bound to the poly-T region of the oligonucleotide followed by ATP hydrolysis-driven unwinding of the duplex region from 3′ to 5′. Once the dsDNA substrate was separated by helicase nanopore, a hairpin formed and the fluorescence of Cy3 was quenched by BHQ2, inducing a fluorescence decrease. This result demonstrated that the helicase nanopore from BPV E1 (306–605, with GST-tag) maintained the capability of unwinding, whereas the BPV E1 (306–605, without GST-tag) and BPV E1 (306–577, with GST-tag) showed no activity (Supplementary Fig. 11). Thus, helicase nanopore from BPV E1 (306–605, with GST-tag) was used in the following studies.

Using linear dsDNA with partially complementary sequence possessing a single strand arm (24-bp dsDNA sequence D, structure shown in Supplementary Fig. 10), we investigated whether the helicase nanopore in BLM retains helicase activity or not. In the presence of dsDNA, blockage signals with a current change of *c.a.* 25 pA can be observed at the voltage of +120 mV (Fig. 3b, buffer condition: PBS, 1.5 mM ATP, 3 mM MgCl₂, pH 7.0). Interesting patterned steps in an individual signal was observed (Fig. 3c–e). They were fundamentally different from the passive ssDNA translocation events driven by voltage, but similar to the single-channel recording signals induced by the topological variations of DNA complex through a solid-state nanopore[43]. A schematic drawing (top right panel in Fig. 3) was proposed according to the structure of the dsDNA and the function of the helicase nanopore, showing the three stages of dsDNA unwinding and translocation through the helicase nanopore corresponding to the three steps in the current recording events: (1) The capture of the 3′ end of the single strand arm by the helicase nanopore; (2) the unwinding process of dsDNA; and (3) the release of ssDNA. In addition, the number of blockage steps could vary from 1 to 3. A faster velocity of dsDNA unwinding was observed at a higher bias voltage for the 24-bp dsDNA as substrates (Fig. 3f), as well as an increased translocation events frequency (Fig. 3g). To investigate the relationship between dsDNA length and unwinding time, the unwinding of dsDNA of different lengths (Sequence E of 44 bp, Sequence F of 80 bp) by the helicase nanopore inserted in BLM under the same voltage of +120 mV were studied (buffer condition: PBS, 1.5 mM ATP, 3 mM MgCl₂, pH 7.0). The results demonstrated that longer unwinding time was required for longer dsDNA (Fig. 3h). The calculated unwinding speed for dsDNA was 7 ± 5 bp per 100 ms under +120 mV. The unidirectionality of dsDNA transport through the helicase nanopore was observed. Whenever dsDNA was added to both *cis*- and *trans*-side of the chamber, the translocation events could only be observed at one voltage direction (Supplementary Fig. 12). Some pores would appear to be inactive while dsDNA was added to one side of chamber. Furthermore, no obvious translocation signals could be observed when the single strand arm was located in 5′ (sequence G of 38-bp dsDNA), thus only the dsDNA with 3′ single strand arm could be transported through the helicase nanopore (Supplementary Fig. 13).

We performed additional experiments to investigate the role of key factors involved in enzyme kinetics, including Mg²⁺ concentration, ATP, and temperature, for dsDNA unwinding through the helicase nanopore. dsDNA (Sequence D of 24 bp, 15 nM) was premixed into the chamber before the helicase nanopore embedded in the BLM (buffer condition: PBS, 1.5 mM ATP, 3 mM MgCl₂, pH 7.0). The blockage signal was hardly observed while the ATP was absent. After addition of EDTA into the buffer solution, the frequency of dsDNA translocation through the helicase nanopore decreased by 90 ± 9% after a short period of time (Fig. 3i, j), which indicated that Mg²⁺ was also required for dsDNA unwinding by the helicase nanopore. For the influence of temperature, dsDNA unwinding frequency decreased by 83 ± 3% at the temperature of 11–14 °C compared with the unwinding frequency at the temperature of 30–34 °C (Fig. 3k, l). Those results were consistent with the enzyme kinetics of BPV E1 in vivo. In our 52 dsDNA unwinding experiments by the helicase nanopore, 25 of them only showed the occurrence of DNA translocation at a positive voltage, while other 27 experiments only showed DNA translocation/unwinding signal at negative voltage. The result indicated there was no directional preference of the helicase nanopore insertion into the BLM in single channel recording (buffer solution: PBS, pH 7.0) (Supplementary Fig. 14).

**Incorporation into live cell membrane and membrane trans-port**. Based on the results of helicase nanopore fusion with vesicle membrane and planar lipid bilayer, we further used patch-clamp

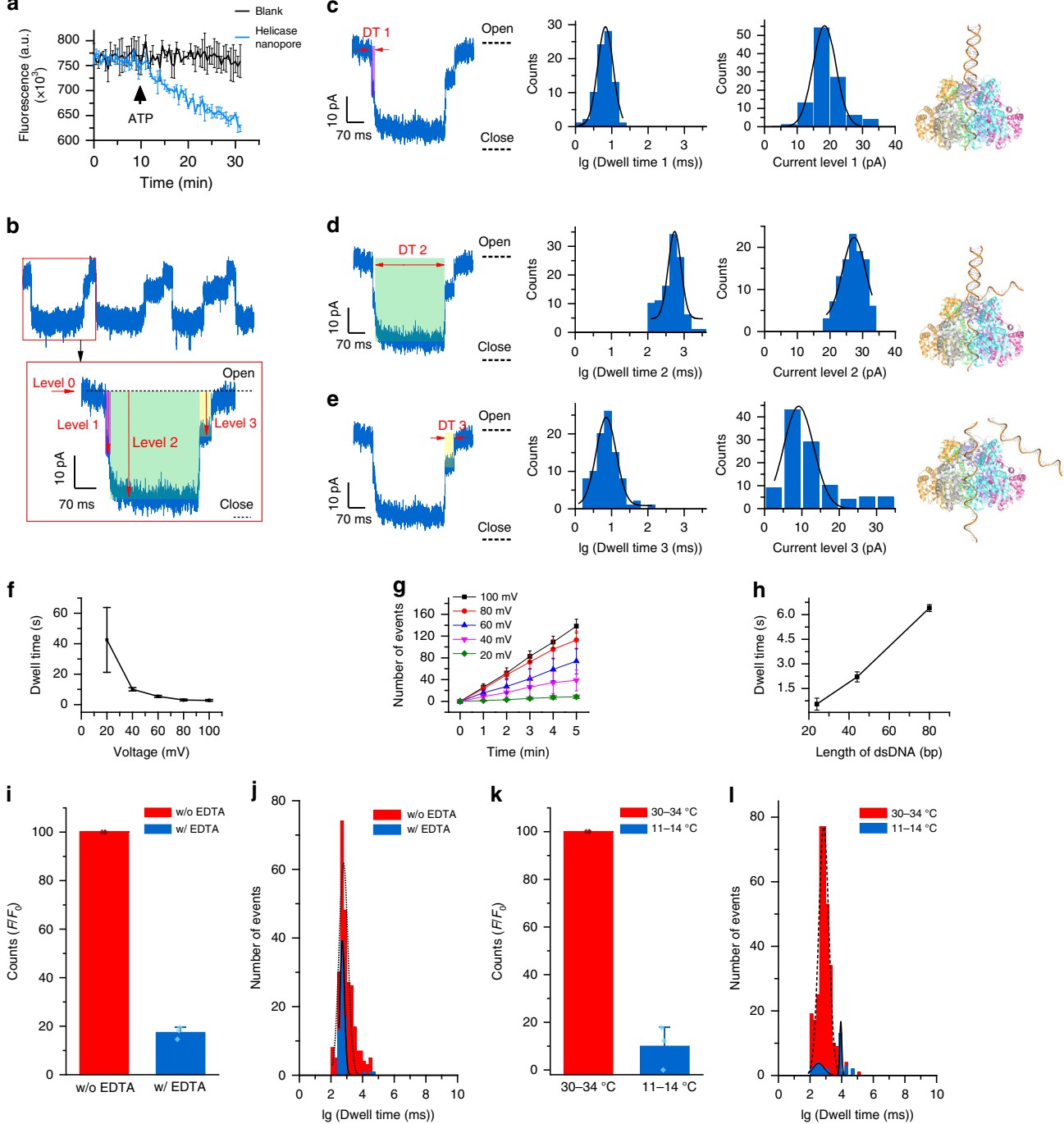

**Fig. 3** Single-molecule dynamics of helicase nanopore unwinding dsDNA. **a** Fluorescence quenching assay showing the unwinding process of dsDNA after the addition of ATP ($n = 3$). **b** Continuous signals caused by dsDNA unwinding through a helicase nanopore embedded in a lipid bilayer membrane (buffer condition: PBS, pH 7.0, under +120 mV, $n = 223$ from 3 trials). **c–e** Typical current stages in a dsDNA unwinding event (left) and the pattern diagram of these three stages (right), with analysis of their current steps and dwell times (middle) (buffer condition: PBS, pH 7.0, under +120 mV **c–e**: $n = 105$). **f** The dependency relationship of unwinding time through the helicase nanopore with bias voltage (24 bp dsDNA) (buffer condition: PBS, pH 7.0, +100 mV ($n = 48$), +80 mV ($n = 75$), +60 mV ($n = 58$), +40 mV ($n = 55$), +20 mV ($n = 16$)). **g** The number of cumulative dsDNA translocation events increased with the elongation of time at different voltage (buffer condition: PBS, pH 7.0, $n = 3$). **h** The unwinding time on the membrane with different lengths of substrate DNA under a bias voltage of +120 mV (buffer condition: PBS, pH 7.0, $n = 3$). **i, j** The number of unwinding and translocation events decreased when the helicase nanopore activities was inhibited by EDTA chelation of $Mg^{2+}$ in electrolyte (buffer condition: PBS, pH 7.0, **i**: $n = 3$; **j**: $n_{(with EDTA)} = 258$, $n_{(without EDTA)} = 72$). **k, l** The number of unwinding and translocation events decreased when the helicase nanopore motor activities inhibited by lower temperatures (buffer conditions: PBS, pH 7.0, $n = 3$; $n_{(30-34°C)} = 333$, $n_{(11-14°C)} = 76$). Error bars in the figure represented the standard deviation between independent experiments. Source data are provided as a Source Data file

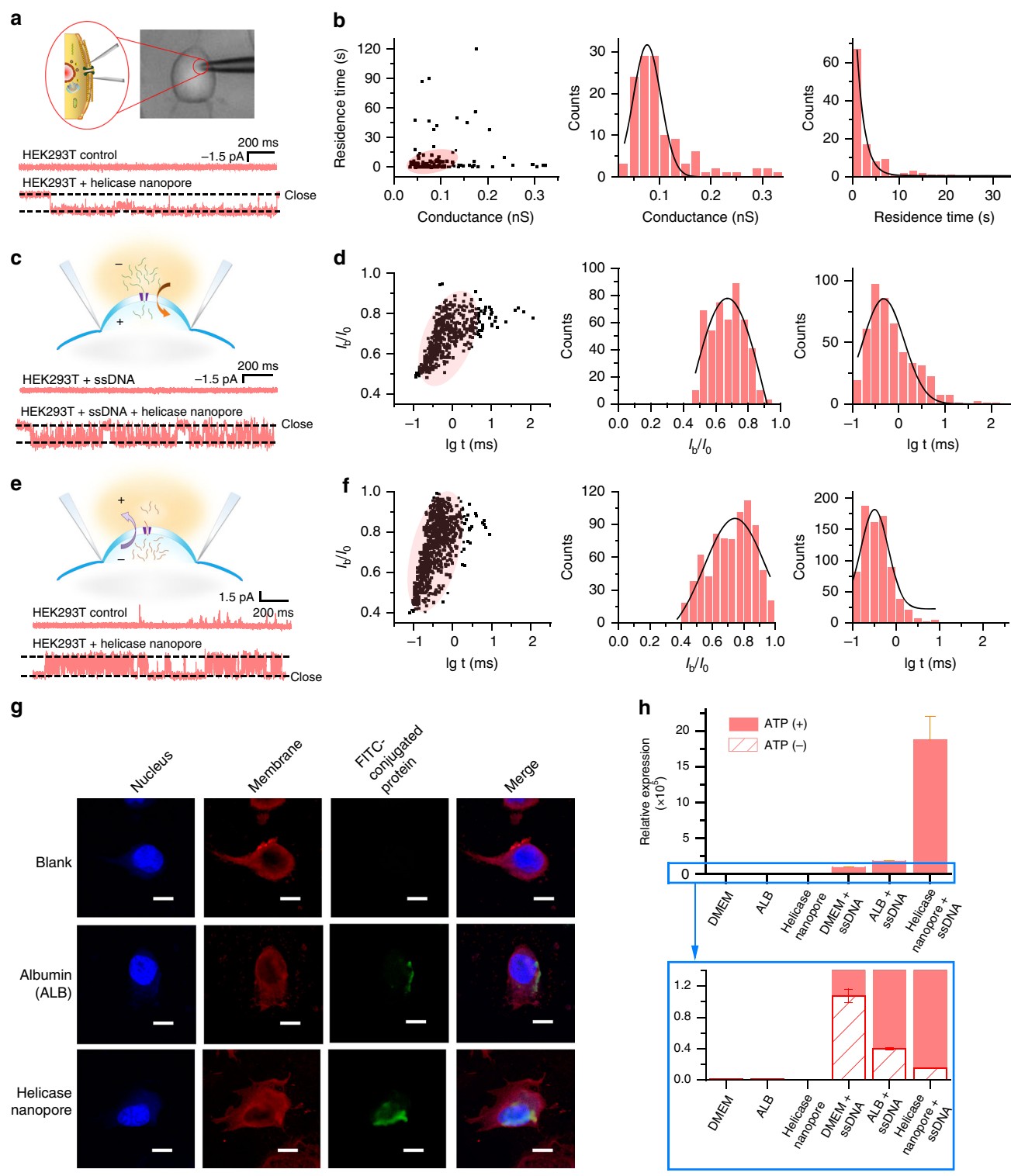

technique to study the ability of the helicase nanopore incorporation into live cell membranes[17]. The human embryonic kidney (HEK293T) cell line was chosen for the patch-clamp experiments since the cell line had low endogenous channel activity and was widely used in cell electrophysiology research. A patch pipette containing the intracellular solution and the helicase nanopores was used to approach a single live cell (Fig. 4a). After gigaseal was formed, resistance current pulses and the resulting changes in electrical conductance of the patch were monitored. In successfully patched HEK293T cells, the cell membrane exhibited

channel-like activities a few minutes after the helicase nanopore addition through the pipette under −60 mV (18 out of 176 cells, Fig. 4a and Supplementary Fig. 15). The mean conductance of the helicase nanopore was $76 \pm 4$ pS (Fig. 4b). Meanwhile, control experiments without helicase nanopore in the patch pipette were performed and showed no apparent channel activity under the same voltage (16 out of 16 cells). Those results confirmed the ability that the helicase nanopore could be spontaneously inserted into live cell membranes. The occurrence of successful incorporation of the helicase nanopore into the cell membrane was *c.a.*

**Fig. 4** Incorporation of the helicase nanopores into live cell membranes. **a** Patch clamp on a HEK293T cell and schematic drawing of a helicase nanopore embedded in HEK293T cell membrane (top), typical current traces of HEK293T cell without (middle) and with (bottom, 70 insertion events from 18 cells out of 176 trials) helicase nanopore under −60 mV. **b** The histogram of the conductance and duration time of current openings by helicase nanopore insertions in HEK293T cell membrane under −60 mV and +20 mV (123 insertion events from 38 cells out of 353 trials). **c** Typical current recording traces of HEK293T cell attached to pipette filled with 200 nM ssDNA solution, without (middle) and with (bottom, 18 insertions from 5 cells out of 98 trials) helicase nanopore under −60 mV. **d** The scattering plot and histogram of the ssDNA translocation through the helicase nanopore in HEK293T cell membrane under −60 mV (475 events from 3 cells). **e** Typical current recording traces of HEK293T cell attached to pipette without (middle) and with (bottom, 14 insertion events from 6 cells out of 79 trials) helicase nanopore under +20 mV. **f** The scattering plot and histogram of the blocking events when a helicase nanopore inserted into the membrane of a HEK293T cell under +20 mV (764 events from 3 cells). Extracellular solution of 145 mM NaCl, 10 mM HEPES, 10 mM glucose, 4 mM KCl, 2 mM CaCl₂, 1 mM MgCl₂, pH 7.4 and intracellular solution of 150 mM KCl, 1 mM EDTA, 10 mM HEPES, pH 7.4, and cell-attach mode were used throughout the experiments in (**a–f**). **g** Confocal laser scanning microscope images of LN-229 cells co-incubated with the helicase nanopore of 0.2 mg/mL for 4 h, scale bar (lower right in the microscopy images) = 10 μm. **h** Q-PCR experiments showing the relative amount of ssDNA translocated into HEK293T cells in the absence and presence of ATP. Q-PCR amplification curves of samples ran in triplicate (*n* = 3). Error bars represented the standard error of mean between three independent experiments. Source data are provided as a Source Data file

10.8% (38 out of 353 cells in total, as shown in Supplementary Fig. 16). Analysis of the insertion was shown in Supplementary Table 3, and the median residence time of the nanopore incorporation on cell membrane was 1.75 s, which was far less than that observed in BLM experiments. These could be attributed to the dynamics of live cell membrane and fusion interaction of the helicase nanopore with the cell.

The translocation of ssDNA through the helicase nanopore in live cell membrane was also studied (Fig. 4c). A glass pipette with premixed helicase nanopore (0.9 μM) and ssDNA (Sequence B of 80 nt, 200 nM) was used for cell patch. After current step corresponding to a pore insertion into the membrane was observed under −60 mV (12 out of 98 trials), large amounts of blockage-like events were observed (5 out of 12 cells, Fig. 4c and Supplementary Fig. 17). The mean blockage was 64 ± 8%, and the mean dwell time of the events was 0.9 ± 0.2 ms (Fig. 4d). As a comparison, no obvious changes in conductance were observed in the control experiment (ssDNA was added into the pipette while the helicase nanopore was not added, 20 out of 20 cells). This confirmed the pipette-assisted real-time, single-molecule voltage-driven delivery of ssDNA into an individual live cell through the helicase nanopore.

The helicase nanopore can also be inserted into the cell membrane under positive voltages (8 out of 79 cells, Fig. 4e and Supplementary Fig. 18) and blockage events were also observed (6 out of 8 cells). The mean current blockage of the events from patched cells was 76 ± 3%, and the mean dwell time for most of the translocation events was 0.5 ± 0.3 ms (Fig. 4f). In contrast to the HEK293T cells control which showed no specific blockage events (top trace in Fig. 4e), transient blockage signals were observed in the same experiment condition for a helicase nanopore embedded in the cell membrane (bottom trace in Fig. 4e). Those control experiments revealed that the signals were associated with the cytoplasm in which water, ion and macromolecules were the major content in cytosol. In the setup of a positive charge applied to the electrode outside the cell membrane, negatively charged macromolecules was expected to transport toward the helicase nanopore from inside the cytoplasm, which could be peptides, RNA, or other small biomolecules, etc. Analysis and statistics of the signals showed a dwell time and blockage percentage comparable to that of ssDNA (Panel d and f in Fig. 4). These results indicated the molecular transport toward the artificial membrane channel formed by helicase nanopore on live cell membrane.

We also carried out the study of spontaneous interaction of the helicase nanopore with live cells. Fluorescein isothiocyanate (FITC)-conjugated helicase nanopore was co-incubated with LN-229 cells for 4 h. FITC-conjugated albumin (ALB) was used as a positive control, as it has been reported that ALB can enter into

cells using gp60[44] mediation. The nuclei were dyed with 4′,6-diamidino-2-phenylindole (DAPI) with blue fluorescence, and the cell membranes were colored by 1,1′-dioctadecyl-3,3,3′,3′-tetramethylindocarbocyanine perchlorate (Dil) with red fluorescence. As shown in Fig. 4g (see Supplementary Fig. 19 for complete view), the green fluorescence emitted under the FITC excitation light[45] demonstrated that the labeled helicase nanopore could be inserted into LN-229 cell membranes and entered the cytoplasm (21 out of 29 cells). To assay the cytotoxicity of the helicase nanopore to cells, the Cell Counting Kit-8 (CCK-8) method was used[46]. During the 72 h observation of the cells co-incubated with helicase nanopore (0.5 mg/mL), the helicase nanopore did not show noticeable cytotoxicity toward LN-229 cells (Supplementary Fig. 20).

In addition to ssDNA translocating through the membrane-adapted helicase nanopore described above, helicase nanopore assisted exogenous ssDNA translocation into live cells in vitro was studied. After incubated with HEK293T cells with both purified helicase nanopore (0.5 mg/mL) and exogenous ssDNA (Sequence B of 80 nt, 110 nM) for 3 h in a cell culture medium which contained extra 2 mM ATP and 5 mM MgCl₂[27], total DNA from the HEK293T cells was extracted for quantitative polymerase chain reaction (Q-PCR). For control groups of DMEM, ALB, and helicase nanopore only, no amplification of ssDNA was observed. When incubated with ssDNA and ATP, helicase nanopore showed higher amplification level of ssDNA compared with DMEM and ALB. (Fig. 4h). Error bars represented the standard error of mean between three independent experiments. In the case of incubation with ssDNA and without ATP, the amplification of ssDNA was much lower and showed no significant difference with DMEM, ALB, and the helicase nanopore (lower panel of Fig. 4h). Those results proved that the ATP was necessary for ssDNA translocation into cells by the helicase nanopore.

## Discussion

Compared with other native membrane channels, the helicase nanopore showed relatively low efficiency of ssDNA internalization into a cell and short residence time at the cell membrane. In order to improve the membrane affinity, further protein engineering is necessary. On the other hand, short residence time on membrane would significantly reduce its toxicity to the cell. Depending on the purpose of application, for example drug delivery, single cell analysis and target therapy to tumor cells, specific interaction could be designed.

In summary, we constructed a helicase nanopore from BPV E1 helicase, and it was reconstituted into lipid bilayer and live cell membranes. It exhibited the ability for passive transport similar to the well-studied protein nanopore, while retaining interesting

helicase activity when embedded in BLM. The dynamics and kinetics of the active transport driven by ATP-hydrolysis was revealed by the current trace recording during individual dsDNA unwinding, providing a powerful tool for single-molecule study of biomotors. The helicase nanopore also formed dynamic channels in live cell membrane, and ssDNA could be delivered into the cell and observed in single-cell level. With the low toxicity to cells, it exhibited potential applications in nanopore sensing, drug delivery and single-cell analysis.

## Methods

**Expression and purification of the helicase nanopore.** The construction of BPV E1 genes with residues 306–577 and 306–605 were designed into pEGX-6P-1 vectors and synthetized by GENEWIZ, Inc. China. Briefly, the E1 306–577 and E1 306–605 plasmids were achieved by PCR. Firstly, E1 306–577 was cloned with primer pair F1 and R1 (Supplementary Table 1) at *BamH*-I and *Xho*-I sites, and E1 306–605 fragment was obtained by addition of E1 578–605 fragment to E1 306–577 fragment by primer pair F2 and R2 (Supplementary Table 1) at *Nsi*-I and *Xho*-I sites. Both two proteins were finally expressed in *E. coli* as N-terminal GST fusion proteins (Supplementary Fig. 2). Using the buffer containing 20 mM Tris and 150 mM NaCl (pH 8.0), the GST fusion proteins were purified by glutathione agarose affinity chromatography (20 mM glutathione added to the buffer) and Superdex™ 200 Increase 10/300 GL column (GE Healthcare). For removal of GST-tag, Prescission Protease (PSP) was added. The uncropped and unprocessed scan of the gel in Fig. 1c is provided as a Source Data file.

**Preparation of helicase nanopore proteoliposome vesicles.** To prepare the giant lipid vesicles, 0.8 mL of 25 mg/mL 1,2-diphytanoyl-sn glycerol-3-phosphocholine (DPhPC) (Avanti, USA) was added to a 2-mL vial. Chloroform was evaporated using a rotatory evaporator. To rehydrate the lipid film, 2 mL of 200–300 mM sucrose was used to bud the vesicles off the glass and into the solution. The vial was then covered with parafilm and stored overnight. Incorporation helicase nanopore into giant vesicles was accomplished as described above, and a volume of 100 μL of helicase nanopore protein were added to the aforementioned dehydrated lipid.

**Cryo-EM specimen preparation and imaging.** The preparation of cryo-EM specimens followed a previous study[47]. In brief, an aliquot (~4 mL) of vesical and the helicase nanopore mixture sample at or ~2.5 mg/mL was placed on a glow-discharged holey carbon film-coated copper grid (QUANTIFOIL® R 2/2, Electron Microscopy Sciences). The samples were blotted with filter paper on both sides at ~100% humidity and 4 °C with a FEI Vitrobot rapid-plunging device and then flash-frozen in liquid ethane. The flash-frozen grids were transferred into liquid nitrogen for storage. Cryo-EM samples were examined using a FEI Talos F200C TEM operating at 200 kV high tension at −178 °C in low-dose mode. A Gatan 626 cryo-holder was used. The micrographs were acquired with a high-sensitivity 4 K × K pixel FEI CETA CMOS camera under a magnification of 22–120 kx (each pixel of micrographs corresponded to ~0.54–0.10 nm of specimen) under a dose of ~10–40 e$^{-1}$/Å$^2$ and a defocus of 1–5 μm. The uncropped and unprocessed of the original cryo-EM images is provided as a Source Data file.

**Incorporation into BLM and electrophysiological assay.** A bilayer of DPhPC was formed on an aperture (200 μm) in a Teflon partition (25 μm thick) (Eastern Scientific LLC, USA) that divided a planar bilayer chamber into *cis* and *trans* compartments (Warner, USA). Planar lipid bilayer was formed using the methods as reported[13]. The aperture (both surfaces) was pre-painted with 0.5 μL 0.5 mg/mL DPhPC n-decane solution twice before loading the buffer, and then painted with 1 μL of 20 mg/mL DPhPC solution in n-decane for bilayer formation. The experiments were performed using a series of symmetrical buffer conditions containing a 2.0-mL solution of various KCl concentrations and 5 mM HEPES (pH 7.5) at 25 °C. The helicase nanopore was added to the *cis* compartment, which was connected to ground electrode and transmembrane voltage was +100 mV. The final concentration of the helicase nanopore was 2–10 ng/mL. Discrete current increase steps could be observed, indicating the incorporation of helicase nanopore into the planar lipid bilayer. Currents were recorded with a patch clamp amplifier (HEKA 10USB, Wiesenstraße 71 D-67466 Lambrecht/Pfalz. Germany, and Axon 200 B, Molecular Devices, USA). They were low-pass filtered with a Bessel filter at 5 kHz and sampled at 100 kHz by a computer equipped with a Patchmaster 2.65.

**ssDNA translocation.** During the ssDNA translocation experiments, ssDNA (Sequence A of 48 nt, TaKaRa, Japan) was added to the *cis*-side or both two chambers, including 5 mM HEPES (pH7.5)/1 M KCl or 0.5 M KCl buffer solution. Two methods were used to add DNA to the chamber for the translocation experiments: (1) DNA was added under a voltage of 0 mV after the nanopore insertion. When the voltage was switched back, the ssDNA moved towards the nanopore via free diffusion of DNA and was driven by bias voltage; (2) DNA was premixed with buffer completely in the chamber before the nanopore insertion.

The DNA movement relied mainly on bias voltage. All ssDNA translocation experiments were performed using the first method.

**Nuclease degradation experiments.** During the ssDNA translocation experiments through the helicase nanopore, 200 nM (final concentration) ssDNA (Sequence B of 80 nt, TakaRa, Japan) was added into the *cis*-chamber, which containing symmetric 5 mM HEPES (pH 7.5)/0.5 M KCl buffer solution. When a large number of blockages signals appeared, adding extra 0.25 μg/mL (final concentration) DNase I into the *cis*-chamber. The experiment was performed at +50 mV at room temperature.

**In vitro helicase activity assay.** In this helicase activity assay, the substrate was prepared by combining single strands at a molar ratio of 1:1 to a final concentration of 50 μM in 20 mM HEPES (pH 7.5), and was placed in 95 °C water bath[48], then allowed to cool to room temperature for ~ 1 h to anneal. The fluorogenic substrate dsDNA (Sequence C of 22 bp) was 5′-CCTAC GCCAC CAGCT CCGTA GG-3′ (5′-Cy5, 3′-BHQ2) and 5′-CCTAC GGAGC TGGTG GCGTA GGTTT TTTTT TTTTT TTTTT TT-3′ (Takara, Japan). Unless otherwise noted, each reaction contained 20 mM HEPES (pH 7.5), 0.7 mg/mL Albumin from bovine serum albumin, 5% Glycerol, 5 mM MgCl$_2$, 3 mM DL-Dithiothreitol (DTT), 5 nM nucleic acid substrate, 50 nM helicase nanopore, and they were initiated with 2 mM ATP[49]. Reactions were carried out in 100 μL, in triplicate, on white half-volume 96-well polystyrene plates at 22 °C. Fluorescence was measured using arbitrary units (a.u.) in each well every 30 s using a 96-well plate. Data were collected using a Microplate Reader Technology Synergy™ 4 equipped with the software Gen5™ Data Analysis Software. Cy5-labeled substrates were measured for excitation/emission at 643/667 nm[42].

**dsDNA unwinding.** During the dsDNA unwinding experiment, PBS (pH 7.0) with 3 mM Mg$^{2+}$ and 1.5 mM ATP was used as the dsDNA translocation buffer. All the dsDNA (Sequence D of 24 bp; Sequence E of 44 bp; Sequence F of 80 bp; Sequence G of 38 bp) were synthesized by Sangon Biotech (Shanghai) Co., Ltd. The dsDNA was prepared by an annealing process: 95 °C for 3 min and the sample was cooled to room temperature slowly for 2 h. Annealed dsDNA was premixed and added into the *trans*-chamber to a final concentration of 15 nM.

**Helicase inhibition assay.** Ethylenediaminetetraacetic acid (EDTA) was used in the inhibition assay of the helicase nanopore. EDTA was added into the chamber while the dsDNA was unwinding through the helicase nanopore. The working concentration of EDTA in the chamber was 1 mM.

**Cell culture.** HEK293T cells (Cell Bank of the Typical Culture Preservation Committee, Chinese Academy of Sciences, Cat. No: SCSP-502) were cultured in high glucose Dulbecco's modified Eagle's medium (DMEM) (Hyclone, USA) with 10% fetal bovine serum (FBS) (Gibco, USA) in a 35-mm culture dish with a 0.8 mm glass slide. The total area of cells could not exceed 30% area of culture dish for follow-up patch clamp experiment. LN-229 cells (ATCC® CRL-2611™) were cultured in DMEM containing 5% FBS[50] in culture flask. All the cells were cultured under 37 °C with 5% CO$_2$ (v/v).

**Cell electrophysiology.** The glass slide containing the cells was set into cell-recording track filled with the following extracellular solution: 145 mM NaCl, 10 mM HEPES, 10 mM glucose, 4 mM KCl, 2 mM CaCl$_2$, 1 mM MgCl$_2$, pH 7.4. The pipettes were produced by a Sutter P-1000 puller from borosilicate glass (Science Products, GB150-10) and filled with intracellular solution: 150 mM KCl, 1 mM EDTA, 10 mM HEPES, pH 7.4; the resistance of the pipette was 6–9 MΩ. The solution containing helicase nanopores was added into the pipette before each experiment, (dilute to 0.054 mg/mL). The concentration of ssDNA (Sequence B of 80 nt) added in the pipette was 200 nM. The data were obtained and preprocessed by the Axon patch-clamp 700B, $V_{hold}$, −60 mV and +20 mV, with a 5-kHz sampling rate, and filtered at 2 kHz.

**Confocal microscopy imaging.** The helicase nanopore was labeled with FITC (FluoroTag™ FITC Conjugation Kit) (Sigma, USA) and the effective tagged helicase nanopore was obtained according to the manufacturer's protocols. The helicase nanopore (0.2 mg/mL) were added into LN-229 cells seeded on Lab-Tek™ II chamber slides (NUNC, USA) and incubated for 4 h. DAPI (Sigma, USA) and Dil (Sigma, USA) were used to label the nuclei and membranes. After sealing the slides with coverslips and antifade agent (ProLong® Diamond Antifade Mountants) (Life, USA), the slides were observed under confocal microscopy (Zeiss, Germany).

**Exogenous ssDNA translocation into cell in cell culture.** For quantitatively analyzing DNA translocation into cells, the Q-PCR method was used. HEK293T cells were incubated with both purified helicase nanopore (0.5 mg/mL) and exogenous ssDNA (Sequence B of 80 nt, 110 nM, Tsingke, China) in DMEM with extra MgCl$_2$ (5 mM) (Sigma, USA) and ATP (2 mM) (Sigma, USA) for 3 h. Total DNA of HEK293T cells was extracted according to the protocols of the

manufacturer (PureLink® Genomic DNA Kits) (Life, USA). Q-PCR was carried out in the CFX Connect Real-Time System (Bio-Rad, USA) and the samples were prepared according to the protocols of the manufacturer (SYBR® Premix Ex Taq™ II) (TaKaRa, Japan). The primers were 5′-CTTCC AGGCA TTAGA GAAAA AGGC-3′ (forward) and 5′-TGTTT TGCGA ACTCC CCAAT ACTT-3′ (reverse).

**Reporting summary**. Further information on research design is available in the Nature Research Reporting Summary linked to this article.

## Data availability

Data supporting the findings of this paper are available from the corresponding authors upon reasonable request. A reporting summary for this Article is available as a Supplementary Information file. The source data underlying Figs. 1c, e, 2b–i, 3a–l, 4a–h and Supplementary Figs. 2c, d, 3, 4, 7, 8, 9, 11, 12, 13, 14, 19, and 20 are provided as a Source Data file (https://figshare.com/s/c87ceb4d356076d0d3a7).

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

## Acknowledgements

This work was supported by the National Key Research and Development Program of China (Grant nos. 2016YFA0201400, 2016YFC1200300, 2017YFC0107603, and 2017ZX10203205-006-002), National Science Foundation of China (Nos. 81541133, 81871448, and 11774279). The authors thank the Instrument Analysis Centre of Xi'an Jiaotong University for cryo-EM imaging.

## Author contributions

J.G. and G.L. conceived the project and designed the experiments with L.Z.'s expertize in cryo-EM imaging. Y.C., S.L., K.S., X. Zeng, X. Zhang, and Changjian Zhao performed the protein expression and purification. Chengjian Zhao and X. Zeng performed the confocal microscopy imaging. L.Z., L.G., and K.S. performed the cryo-EM characterization. X.J., R.J., X.L., L.D., and L.W. performed the cell patch clamp experiments. K.S., Changjian Zhao, H.N., M.Z. and R.T. performed the single channel recording and ssDNA translocation experiments. Changjian Zhao performed the dsDNA unwinding experiments. X.D. and H.X. contributed the single-cell analysis experiment design. B.S., X.W., and B.Y. contributed to the data analysis of cell study. C. Zhou and Y.H. contributed to the data analysis of single channel recordings. J.G. and G.L. wrote the paper, and K.S., Changjian Zhao, Y.C., X. Zeng, X.J., E.S., Q.Y., and P.C. assisted the paper preparation. All authors discussed the results and commented on the paper.

## Competing interests

The authors declare no competing interests.
