## [Peer Review File · Nature Communications]

Reviewers' Comments:

Reviewer #1:

Remarks to the Author:

In the manuscript by Sun et al, the authors inserted BPV E1 helicase into the artificial lipid bilayer and cell membrane and translocated ssDNA or dsDNA.

Overall, the results are interesting, and the paper is well written. However, I have a few questions related to technical rigor and data presentation clarity in the manuscript, which need to be addressed. My comments are below:

1) the IV curve in Figure 2d is a bit peculiar, why does this pore show spikes for both directions of applied voltage, in the range 50 to 100 mV and -50 to -100 mV ?

2) The voltage applied for Figure 2e is not described in the caption.

3) The ssDNA events shown in panel e, and their analysis in panels f and h, indicate that only a minor fraction of the current is blocked by the pore. In panel e it seems that ~15-20 pA amplitudes are obtained, and based on the pore conductance the 1.3 nS conductance, the baseline current level should be ~65 pA at 50 mV. Can the authors explain the current blockade histograms in panels f and h? What is "I" here, the current blocked, or the current level?

4) The capture rate of ssDNA is extremely peculiar. The authors obtain tens of events per seconds for 25 nM ssDNA at (I presume) a weak voltage. For comparison, the capture rates for hemolysin at >100 mV is ~1-2 orders of magnitude smaller. I would like to see a summary of rate vs. concentration.

5) The degradation experiments in Figure S5 are interesting. What are the deep spikes obtained for the pore (both before and after DNA degradation)? They seem to be DNA independent spikes, but correspond more to the expected blockade level for DNA than the shallower spikes.

6) Why is the baseline current in experiments shown in the supplemental information (Fig S6, S7 for example) so noisy as compared with the data in Figure 2? Is there something fundamentally different about the experiment?

Reviewer #2:

Remarks to the Author:

The paper by Sun et al. is one of the more interesting in the nanopore field, in recent years.

Its main weak point is the lack of precise description. This could be largely remedied by providing for each experiment:

- the salt concentration
- the applied potential (use + for a positive potential: +100 mV not 100 mV)
- pH value
- the helicase used

A confusing collection of descriptors is used: channel, connector, helicase, nanopore, E1, fusion protein. Just say which helicase was used and whether or not the GST region was cleaved

- show $i = 0$, the zero current level in all traces
 - name the DNA used and display the DNA structures in the paper
- the nature of the dsDNA is confusing as it does not seem to be entirely double-stranded
- where appropriate indicate the number of times an experiment was repeated for each datapoint

Further, the meaning of I in I/I₀ should be clarified. Earlier there is $I_r = I_o - I_b$, I is probably I_b

It would be useful to tabulate all the I/I₀ and tau values mentioned in the paper and the conditions under which they were obtained. Then, they might be compared.

Additional points:

p1: The authors state that DNA has been driven through pores by "passive transport", but of course enzymes have been used previously. The first example was [1] and later published examples include work by the Akeson and Gundlach groups. Of course, Oxford Nanopore use an enzyme-driven process.

p1: The authors state that optical tweezers etc have been used to study DNA-handling enzymes, but again nanopores have been used: e.g. [2].

p1/ 2: The authors state that the E1 helicase was engineered so that it would insert into membranes, but they just seem to use truncated forms of the protein. Normally proteins insert into membranes because they have hydrophobic surfaces. It would be useful if Fig 1a, b, d showed hydrophobic residues, e.g. in green.

The cryoEM images (Fig 1 e, f) are unconvincing. It is not even clear which objects are considered to be the helicase by the authors, and which part of it is considered to be in the bilayer.

It seems more likely that the helicase sits on the surface of the bilayer in the manner described for truncated α -hemolysin pores [3]. [3] is an important paper in this area and it should be cited and the examples in it should be mentioned.

p2: It is not clear what the percentage insertion into bilayers means or how it was obtained. Why would a GST tag increase the efficiency of insertion?

The conductance calculations on p3 and elsewhere are not valid because the pore lumens are not cylindrical.

p3 and elsewhere: it is not strictly proven that the DNA is translocated through helicase. It might just bind and dissociate.

p4 (and also p5): The authors should clearly describe which ends of the ssDNA and dsDNA enter the pore and what they mean by 3' to 5' polarity.

In Fig 2a dsDNA enters the wide end of the helicase. In Fig 3 cde, it enters the narrow end

p5: It is unexpected that the helicase can insert into a bilayer in either orientation. Do the authors have an explanation.

E1 506-577 has no unwinding activity--- does it transport ss or ds? --- the authors need to be clear which E1 they are talking about at all points in the paper.

p4: What does linear dsDNA with sticky ends mean here?-- show the constructs used in the paper

p4: The patterns seen for the interaction with dsDNA should be clearly described in the text and some attempt made to interpret them. The observations in reference 32 should also be clearly explained-- why should they be similar to what is seen in this paper? They do not seem to be relevant.

The translocation rate of DNAs into cells should be quantified and compared with the event rate in

planar bilayers.

p6 top: 76.3 pS-- what 'channel' is this?

p6: "the blockage signals represented the molecular transport extracted from inside the cell" What does this mean? What was transported?

p7: "genomic DNA" -- what is intended here?

p7: ssDNAs transported into cells--- which DNA used?

p9/ 10: explain how helicase and unwinding activities differ

p11: What do "transfection" and "genomic DNA" mean here?

Fig 1d-- what is the evidence for the structure shown? Where are the hydrophobic residues on the helicase? Is the interaction demonstrated by Stoddart more likely [3].

Fig 1ef- Put e and f on the same scale. Indicate the structures that are thought to be helicase molecules.

Fig 2-- Why are b and c at different salt concentrations?

Fig 2 e--- Show an expanded segment so that the nature of individual blockades can be discerned

Fig 3: show the zero current level on each trace

Fig 3: Step 3 (in b- e) seems longer than Step 1. This is not the case in the histograms

Fig 3: The nature of the events underlying the current levels is proposed in the structures in the top right--- these ideas are not discussed in the text.

The origin of the blockades in Fig 4e is unclear

There is no abstract

These are remarkable results and the editors should ensure that the authors make their constructs available so that the results can be duplicated by others.

References

1. Cockroft SL, Chu J, Amarin M, Ghadiri MR: A single-molecule nanopore device detects DNA polymerase activity with single-nucleotide resolution. *J Am Chem Soc* 2008, 130:818-820.
2. Derrington IM, Craig JM, Stava E, Laszlo AH, Ross BC, Brinkerhoff H, Nova IC, Doering K, Tickman BI, Ronaghi M, et al.: Subangstrom single-molecule measurements of motor proteins using a nanopore. *Nat Biotechnol* 2015, 33:1073-1075.
3. Stoddart D, Ayub M, Höfler L, Raychaudhuri P, Klingelhoefer JW, Maglia G, Heron A, Bayley H: Functional truncated membrane pores. *Proc. Natl. Acad. Sci. USA* 2014, 111:2425-2430.

Reviewer #3:

Remarks to the Author:

In their work, Sun et al report that the E1 helicase from Bovine Papilloma Virus can be inserted

into lipid membranes and be used as a nanopore to translocate ssDNA. The authors performed a series of in vitro and in cell electrophysiological experiments to prove the activity of the inserted helicase. Remarkably, they show that under certain conditions ssDNA translocation can be performed in an active manner, fueled by ATP hydrolysis. This is a significant advance in the field, which can be of potential interest for the community. However, there are several important issues which should be addressed before considering the work for publication.

Main points:

1. Helicase incorporation and engineering:

Helicase incorporation into the membrane is not directed and non-specific, which impedes the authors to embed the helicase in the membrane in a preferred orientation and favors short residence times (time helicase bound to membrane). Taken together, these limitations hinder the possible applications of this system (see also comments about statistics below). The authors may comment.

The authors should explain briefly why BPV E1 306-577 and 306-605 fragments were chosen and show how deletions affect the in vitro activity of the enzyme. This is not shown.

In order to increase the solubility of the protein, the authors fused a GST-tag to the above constructs. However, it is not stated which constructs were used in each of the experiments shown in the manuscript. The effect of the GST-tag in the in vitro helicase activity is also not reported.

The putative role of lipid composition on helicase insertion into GUVs is not explored nor discussed.

What is the average residence time of the nanopore in GUVs or 'synthetic' membranes?

Cryo-EM images are not conclusive and hard to see. Same magnification should be used in control and no-control images.

2. ssDNA translocation: Duplication of bias voltage (+50 to +100 mV) increased the frequency of ssDNA translocation events a merely 1.5%, from 81 to 82.5%. Of note is that at 100 mV the distribution (figure 2h) is broader, implying higher error. So, it seems there is no significant correlation between the number of events and voltage, which is odd; however the authors stated this is a significant increase, why is that?.

3. dsDNA unwinding experiments:

The authors stated that 3 steps are observed during dsDNA unwinding by the helicase. However, at least 4 different steps are clearly distinguishable in each trace (an additional step is clearly visible after step 3). The authors may comment on the nature of these steps. How do the properties of each step depend on ATP concentration, temperature, DNA length, etc? What is the variability of these steps from one trace to another?

Overall, the work will increase in scope if the authors prove that their method is robust enough for detailed biophysical characterization of the helicase activity.

The authors reported an average dsDNA unwinding rate of 64.54 +/- 48.06 bp/s. So, the average error seems to be ~75%. However, the data presented in figure 3h (duration time vs. length of dsDNA) presents very small error bars, suggesting very homogeneous unwinding rates between different nanopores. How is this possible? Also, data in figure 3h indicate an average unwinding velocity of ~13-16 bp/s, which is ~4 times slower than the reported average dsDNA unwinding rate (64.54bp/s). On the other hand, how do these rates compare with unwinding rates in vitro?

Importantly, the statistics (number of unwinding events) in each experimental condition are not

indicated. The authors should indicate clearly the statistic in all conditions.

Errors bars are not defined (both for figure 3 and figure 4). Do they show s.d., s.e.m.?

There is some confusion about dwell times (figure 3f), duration times (figure 3h) and average velocities. The authors should use uniform terminology and clarify these points. Figure 3f shows dwell times vs voltage, but for what DNA length? Figure 3h shows duration time vs length of dsDNA, but for which voltage?

The temperatures reported in the main text are different from the temperatures indicated in the plots (figure 3i and 3j). According to the main text, Figures 3i and 3j should correspond to figures 3k and 3l (and the other way around).

The authors should comment on and elucidate all these points carefully. As mentioned, proper/rigorous biophysical characterization of the nanopore is key to validating their method. Overall, the presentation of the biophysical characterization of the nanopore is very confusing.

4. Incorporation and activity measurements of helicase in cell membranes:

The reported efficiency of the method to internalize ssDNA into cells using the helicase-nanopore is ~5% (5 out of 98 cells internalized ssDNA upon membrane modification with the nanopore) and the residence time of the helicase at the cell membrane is short (not specified). These are important drawbacks of the method that will limit its possible applications in the future. The authors may comment.

The number of control experiments seems not sufficient. Only 8 cells were used as controls. Considering that the efficiency of the method is ~5%, controls should include more than 8 cells.

What is the actual residence time of the nanopore in cell membranes? The authors stated in the main text that it is less than 10 seconds, the figures show time windows < 2 seconds.

The authors suggested the possibility of transport of intracellular material outside the cell by the nanopore when a positive voltage is applied (page 6, line 172). How do the authors know that the observed blockage events correspond to transport events? Could it just be that the pore gets clogged with intracellular material migrating to membrane due to voltage application?

The procedure to determine ssDNA translocation into cells using ATP (figure 4h) and its validation are hardly described.

Overall, in its current state, the work seems a collection of experiments; accurate, meticulous characterization of the nanopore in each condition is missing. The manuscript will benefit greatly from English editing, reorganization and a more detailed explanation of experimental conditions in each case.

Response to the Reviewers comments

The comments and suggestions made by the reviewer are very helpful for us to revise the manuscript. A point-by-point response to the comments is listed below.

Reviewer #1

In the manuscript by Sun et al, the authors inserted BPV E1 helicase into the artificial lipid bilayer and cell membrane and translocated ssDNA or dsDNA.

Overall, the results are interesting, and the paper is well written. However, I have a few questions related to technical rigor and data presentation clarity in the manuscript, which need to be addressed. My comments are below:

Response: We appreciate the reviewer's time and expertise to evaluate our work, along with the constructive comments and suggestions.

1) The IV curve in Figure 2d is a bit peculiar, why does this pore show spikes for both directions of applied voltage, in the range 50 to 100 mV and -50 to -100 mV?

Response: To investigate the possible cause of the spikes, we performed additional experiments and analysis of the single channel recordings of helicase nanopores under ramping voltage. Out of the 100 independent trials, 93 showed no spikes while 7 showed stochastic spikes (as shown in Figure R1). This revealed that the spikes were attributed to occasional background noise rather than specific transport signals. We also updated the Figure 2d in the revised manuscript.

Figure R1. The representative current traces of the helicase nanopore under ramping potential and

statistics of the recording. Recording conditions: a, b, e, f (1 M KCl, from -200 mV to +200 mV); c, d, g (0.5 M KCl, from -200 mV to +200 mV); h, i, j, k (0.5 M KCl, from -100 mV to +100 mV); l, Histogram of the recordings with and without spikes. (n=100)

2) The voltage applied for Figure 2e is not described in the caption.

Response: The voltage applied for Figure 2e was -50 mV, and the caption was updated as below:
e. Typical current traces of blockages caused by 100 nM ssDNA (Sequence A), and the control without ssDNA added. (0.5 M KCl, 5 mM HEPES, pH 7.5, - 50 mV, n≥3).

3) The ssDNA events shown in panel e, and their analysis in panels f and h, indicate that only a minor fraction of the current is blocked by the pore. In panel e it seems that ~15-20 pA amplitudes are obtained, and based on the pore conductance the 1.3 nS conductance, the baseline current level should be ~65 pA at 50 mV. Can the authors explain the current blockade histograms in panels f and h? What is "I" here, the current blocked, or the current level?

Response: The electrolyte conditions in Figure 2 panel e was 0.5 M KCl, same with that in panel f, g, h and i. The conductance of the helicase nanopore under this condition was 0.556 ± 0.11 nS in the revised manuscript), and 80% blockage of the pore would result in amplitudes of ~22 pA. This coincide with the value in the blockage histogram in panel f and h. We have updated the figure captions with full description of electrolyte buffer and bias voltage for clarity.

As suggested by the reviewer, we use "I_b" to describe the current blocked, and "I₀" as the opening channel level. All the related figures and text were updated in the revised manuscript.

4) The capture rate of ssDNA is extremely peculiar. The authors obtain tens of events per seconds for 25 nM ssDNA at (I presume) a weak voltage. For comparison, the capture rates for hemolysin at >100 mV is ~1-2 orders of magnitude smaller. I would like to see a summary of rate vs. concentration.

Response: Following the reviewer's suggestions, we performed additional ssDNA (Sequence A) translocation experiments at 5 different concentrations (25 pM, 250 pM, 10 nM, 20 nM and 25 nM. n>3 for each concentration), and a linear relationship could be observed between the translocation events frequency vs. the ssDNA concentration (Figure R2 a, b, d / S9).

The reason for faster DNA capture rates by the helicase nanopore could be attributed to the nature of the protein. Compared with bacteria derived pore-forming toxin hemolysin or porin MspA nanopores, the virus-derived helicase motor process and unwind dsDNA in host cell during infection. Those summary and discussions were included in the revised manuscript.

Figure R2. (S9). The correlation between ssDNA translocation events frequency and concentration. a, Representative current traces at the concentration of 25 pM, 250 pM and 20 nM ssDNA respectively (0.5 M KCl, pH 7.5, bias voltage of +100 mV, $n \geq 3$); b and d, The relationship between concentration of ssDNA and translocation events frequency (0.5 M KCl, bias voltage of +50 mV, $n \geq 3$); c, The relationship between the translocation events frequency and bias voltage (0.5 M KCl, 250 pM ssDNA, $n \geq 3$); e, The relationship between the interevent gap time and bias voltage (0.5 M KCl, 250 pM ssDNA, $n \geq 3$).

5) The degradation experiments in Figure S5 are interesting. What are the deep spikes obtained for the pore (both before and after DNA degradation)? They seem to be DNA independent spikes, but correspond more to the expected blockade level for DNA than the shallower spikes.

Response: The signals of deep spikes from current traces before DNA degradation with sharp triangle shape were shown in magnified view (in blue boxes) in Figure R3. They resemble the deep spikes obtained after DNA degradation. In comparison with the blockage signals brought by ssDNA translocation (in red boxes), we agree that they were DNA independent spikes which could be attributed to non-specific background signals.

Figure R3. Single channel recording of enzymatic degradation of ssDNA by DNase I. a. A typical current trace of adding ssDNA into the cis chamber after helicase nanopore insertion. b. The current trace of ssDNA after addition of 0.25 $\mu\text{g/mL}$ DNase I into the cis chamber after 20 min. 200 nM ssDNA (sequence C) was used in 5 mM HEPES, pH7.5, 0.5 M KCl electrolyte buffer with + 50 mV applied potential.

6) Why is the baseline current in experiments shown in the supplemental information (Fig S6, S7 for example) so noisy as compared with the data in Figure 2? Is there something fundamentally different about the experiment?

Response: Baseline current in Figure 2 were *c.a.* ± 1.7 pA for helicase nanopore incorporation and ssDNA translocation, while unwinding experiment in Figure S12 and S13 showed a baseline

current of *c.a.* ± 3.9 pA. The major difference in experiment conditions were the additional ATP and Mg^{2+} in the latter. Thus the higher baseline level could be introduced by those components, and the unwinding dynamics during helicase function.

Reviewer #2 (Remarks to the Author):

1) The paper by Sun et al. is one of the more interesting in the nanopore field, in recent years.

Response: We appreciate the insightful comment and constructive suggestions from the reviewer.

2) Its main weak point is the lack of precise description. This could be largely remedied by providing for each experiment:

- the salt concentration
- the applied potential (use + for a positive potential: +100 mV not 100 mV)
- pH value
- the helicase used

Response: We have completed all the experiment detail descriptions in the revised manuscript including the salt concentration, applied potential, pH value, number of experiments et al., and specified the type of helicase used in each experiment where applicable.

3) A confusing collection of descriptors is used: channel, connector, helicase, nanopore, E1, fusion protein. Just say which helicase was used and whether or not the GST region was cleaved

Response: Following the suggestion, we unify the terminology as helicase nanopore in the revised manuscript. The truncated BPV E1 (306-605) helicase with GST-tag were used through the study. The comparison between the helicase with and without GST-tag were presented in Figure S5.

4) Show $i = 0$, the zero current level in all traces

Response: Since both positive and negative bias voltages were used for helicase nanopore experiments in BLM and cell, we added baseline current level (close status) and opening level in all traces where applicable. Source data used to generate the traces can be found in the Source Data file provided online.

5) Name the DNA used and display the DNA structures in the paper

Table R1 (Table S2) List of DNA sequences used in the study

Description	Sequence	Length
Sequence A	5'-(T) ₉ CCCCCCTTTTTGGGGGTTTTTAAAAA(T) ₉ -3'	48 nt
Sequence B	5'-TGTTTTGCGAACTCCCCAATACTTTTCTTTTCAAATTTAAAATCTGCTCCTCACC CGCCTTTTTCTCTAATGCCTGGAAG-3'	80 nt
Sequence C	5'-/Cy5/CCTACGCCACCAGCTCCGTAGG/BHQ2/-3' 3'-(T) ₂₀ GGATGCGGTGGTTCGAGGCATCC -5'	22 bp
Sequence D	5'-AGCTCCACCCCTCCTGGTAACCAG(T) ₂₀ -3' 3'-TCGAGGTGGGGAGGACCATTGGTC(T) ₂₀ -5'	24 bp
Sequence E	5'- CAGAGGACAGATAGGGCGGGTGCAAACCTTTCGCGGGGAGCAGCC(T) ₆ -3' 3'- GTCTCCTGTCTATCCCGCCACGTTTGAAAGCGCCCCTCGTCG-5'	44 bp
Sequence F	5'-CAGGCAGAGGACAGATATTTGACTTGCATAGTCGGGTGCAAACCTTTCGCTTTG AGCAGCCATGCACAGATGAATCGGG(T) ₂₀ -3' 3'-GTCCGTCTCCTGTCTATAAACATGAACGTATCAGCCCACGTTTGAAAGC GAAACTCGTCGGTACGTGTCTACTTAGCCC-5'	80 bp
Sequence G	5'- ACAGAATAGGGCTAACAGACAAGAGGCATA AACAGGGTAGGGTACGGGAA- 3' 3'- ATTGTCTGTTCTCCGATTTGTCCCATCCCATGCCCTT- 5'	38 bp

Figure R4 (S11). The structures of dsDNA used in the study.

Response: We summarized and named the DNA sequence used in the study in Table R1 (Table S2). The structures of the dsDNA were predicted by NUPACK (<http://nupack.org>) and displayed in Figure R4 (S11). Related description and figure legends were updated in the revised the manuscript.

6) The nature of the dsDNA is confusing as it does not seem to be entirely double-stranded.

Response: That's correct. dsDNA with sticky end was used for BPV E1 helicase itself to assist non-specific unwinding *in vitro*^{1,2}, and we designed the dsDNA (sequence E) with sticky end (structure displayed in Figure R4) in unwinding experiments for easier capture of the dsDNA in this study.

7) Where appropriate indicate the number of times an experiment was repeated for each datapoint

Response: Descriptions about the number of times of all experiment were added where applicable in the revised manuscript text and figures.

8) Further, the meaning of I in I/I₀ should be clarified. Earlier there is I_r = I₀ - I_b, I is probably I_b

Response: We defined and updated those parameter (I₀, I_b, and I_r) throughout the revised manuscript following the suggestion.

9) It would be useful to tabulate all the I/I₀ and tau values mentioned in the paper and the conditions under which they were obtained. Then, they might be compared.

Response: We calculated and listed the peak I_b/I₀ and tau values in Table R2. It can be observed that under the same buffer condition, increasing bias voltage would result a decreased τ_{on} and τ_{off} .

Table R2 I_b/I₀ and tau values for the helicase nanopore in different experiment conditions

I _b /I ₀	Log (τ_{on}) / (ms)	τ_{off} / ms	Voltage / mV	Buffer	
81.85 ± 0.007%	3.49 ± 0.035	1.11 ± 0.16	50	0.5 M KCl	Figure 2f, g
78.3 ± 0.017	3.324 ± 0.034	0.88 ± 0.09	60	0.5 M KCl	Figure R2
80.11 ± 0.021	3.15 ± 0.025	0.75 ± 0.06	70	0.5 M KCl	Figure R2
80.5 ± 0.459%	2.12 ± 0.036	0.636 ± 0.02	100	0.5 M KCl	Figure 2h, i

10) The authors state that DNA has been driven through pores by "passive transport", but of course enzymes have been used previously. The first example was and later published examples include work by the Akeson and Gundlach groups. Of course, Oxford Nanopore use an enzyme-driven process.

[1] Cockroft SL, Chu J, Amarin M, Ghadiri MR: A single-molecule nanopore device detects DNA polymerase activity with single-nucleotide resolution. *J Am Chem Soc* 2008, 130:818-820. The authors state that optical tweezers etc have been used to study DNA-handling enzymes, but again nanopores have been used: e.g. [2].

[2] Derrington IM, Craig JM, Stava E, Laszlo AH, Ross BC, Brinkerhoff H, Nova IC, Doering K, Tickman BI, Ronaghi M, et al.: Subangstrom single-molecule measurements of motor proteins using a nanopore. *Nat Biotechnol* 2015, 33:1073-1075.

Response: We thank the reviewer for this suggestion, and introduced those pioneering work in the revised manuscript as below.

Original: However, most translocation through these nanopores is in the form of passive transport driven by an external force such as voltage, and the nanopore functions are primarily studied in vitro.

Revised: However, translocation of DNA through these nanopores itself are usually driven by an external force such as voltage, and the nanopore functions are primarily studied in vitro. Enzymes assisted unwinding of dsDNA through nanopore is realized^{3,4} and designed for DNA sequencing applications ingeniously⁵.

Original: Single-molecule studies provide a powerful tool revealing the detailed dynamics of biomotors including the DNA unwinding process by helicases, such as optical tweezers, Fluorescence resonance energy transfer (FRET) microscopy, and DNA curtains.

Revised: Single-molecule studies provide a powerful tool revealing the detailed dynamics of biomotors including the DNA unwinding process by helicases, such as optical tweezers, Fluorescence resonance energy transfer (FRET) microscopy, nanopore⁶, and DNA curtains.

11) The authors state that the E1 helicase was engineered so that it would insert into membranes, but they just seem to use truncated forms of the protein. Normally proteins insert into membranes because they have hydrophobic surfaces. It would be useful if Fig 1a, b, d showed hydrophobic residues, e.g. in green.

Response: We prepared Figure R5 according to the suggestion, showing hydrophobic residues in green. It can be observed that the more hydrophobic residues (green) spread over the middle of the helicase nanopore facing the N-terminal (panel b left), which could assist the insertion into the

lipid bilayer.

Figure R5. (S1). Illustration of the helicase nanopore structure showing hydrophobic residues. a, Side view; b. Bottom view; c. Schematic drawing of a helicase nanopore in a lipid bilayer. According to the analysis of membrane insertion results, the helicase nanopore can be inserted into lipid bilayer membrane in either orientation. N-terminal insertion case was shown in the illustration. All structural hydrophilic residues are colored in white, with the hydrophobic residues in green.

12) The cryoEM images (Fig 1 e) are unconvincing. It is not even clear which objects are considered to be the helicase by the authors, and which part of it is considered to be in the bilayer.

Figure R6. (Figure S3). The Cryo-EM images of proteoliposomes vesicles containing helicase nanopores (8 from 45 samples). Inset, overlaid outlines of the bilayer (red solid lines) and helicase nanopore (blue dashed lines).

Figure R7. (S4). The statistical results of helicase nanopore insertion into vesicles ($n=67$). a. Radius plot of the histogram of helicase nanopore tilt angles measured relative to the axis normal to the bilayer plane. b. Histogram of location of the helicase nanopore to the with vesicles. c. Histogram of the insertion part of helicase nanopore in the membrane. d. Scatter plot of the thickness of lipid membrane vs the inserted length (magnified view shown in box). e. Histogram of the insertion orientation towards the bilayer.

Response: To improve the quality of cryo-EM images of helicase nanopore proteoliposomes vesicles, we conducted extensive amount of additional cryo-EM characterization of the vesicles containing helicase nanopores. High resolution images of 67 helicase nanopores from 45 images were collected and analyzed (representatives images shown in Figure 1e, R6), and the data set of original cryo-EM images of the 45 samples were provided. Red solid lines and blue dashed lines highlighting the bilayer membrane and helicase nanopore respectively were provided in the insets. Further detailed analysis was performed on the 67 membrane-embedded helicase nanopores in the following aspects (Figure R7):

- Helicase nanopore tilt angles measured relative to the axis normal to the bilayer plane;
- Location of the helicase nanopore to the vesicles: inside or outside;
- Interaction with the bilayer: transmembrane insertion or surface attachment;
- Relationship of inserted length vs. the thickness of the bilayer;
- Orientation of the helicase nanopore towards the bilayer membrane.

Figures, description and discussion of those images and analysis were included in the revised manuscript.

13) It seems more likely that the helicase sits on the surface of the bilayer in the manner described for truncated α -hemolysin pores [3]. [3] is an important paper in this area and it should be cited and the examples in it should be mentioned.

[3] Stoddart D, Ayub M, Höfler L, Raychaudhuri P, Klingelhoefer JW, Maglia G, Heron A, Bayley H: Functional truncated membrane pores. *Proc. Natl. Acad. Sci. USA* 2014, 111:2425-2430.

Response: We compared and discussed the pioneering work⁷ in the revised manuscript. From panel c and d in Figure R7, it can be concluded that about half of the helicase nanopores were surface attached on the bilayer (similar to the truncated α -hemolysin pores), and another half were inserted into the bilayer with varying depths.

14) It is not clear what the percentage insertion into bilayers means or how it was obtained. Why would a GST tag increase the efficiency of insertion?

Response: Statistics in Figure S6 revealed the efficiency of helicase nanopore insertion into the bilayer, which was calculated by ratio of numbers of observed insertions to the total number of trials in single channel recording experiments. Corresponding descriptions were updated in the revision to avoid confusion.

GST tag is a soluble fusion protein, which stabilize the helicase nanopore in the solution. The helicase nanopore without GST tag tends to aggregate, thus the GST tag increase the efficiency of insertion in an indirect manner. Those discussions were added to the revised manuscript.

15) The conductance calculations on p3 and elsewhere are not valid because the pore lumens are not cylindrical.

Response: We have modified the calculations of conductance of helicase nanopore using the equation from Hall⁸. The equation was used on the alpha-Hederin nanopore sharing similar lumen characteristics with the helicase nanopore. The variation of the conductance as a function of the diameter can be described as below:

$$G_{nanopore} = \sigma_{KCl} \left(\frac{4h}{\pi d^2} + \frac{1}{d} \right)^{-1}$$

where σ_{KCl} (11.1 S/m) is the molar conductivity of the electrolyte (1 M KCl), h is the effective thickness of the nanopore, and d is the diameter of the nanopore. Effective mean thicknesses of 6.7 nm and the diameter range from 1.1-1.5 nm for helicase nanopore were obtained from its crystal structure^{9,10,11}. The calculated electric conductance for the helicase nanopore in 1 M KCl solution (electrolyte conductivity 11.1 S/m) was 1.39-2.49 nS. This is comparable to the measured conductance value (1.34 ± 0.02 nS). We updated the calculation in the revised manuscript as well.

16) and elsewhere: it is not strictly proven that the DNA is translocated through helicase. It might just bind and dissociate.

Response: To address this question, we carried out additional concentration-dependent ssDNA translocation experiments. As shown in Figure R8 (Fig. S9), with ssDNA concentration increased from 25 pM to 25 nM, the translocation events frequency showed linear increase (panel a). Moreover, with the bias voltage increased from +50 mV to +100 mV, the dwell time of ssDNA translocation reduced significantly (panel b). These data demonstrate that the blockage signals are associated with ssDNA translocation, rather than random collision.

Figure R8. (S9). Analysis of concentration and voltage dependent of ssDNA blockage events. a. ssDNA blockage events frequency vs. DNA concentration (0.5 M KCl, 5 mM HEPES, pH 7.5, +50 mV). b. Dwell time of blockage vs. bias voltage (0.5 M KCl, 5 mM HEPES, pH 7.5, 250 pM ssDNA).

17) The authors should clearly describe which ends of the ssDNA and dsDNA enter the pore and what they mean by 3' to 5' polarity.

Response: It was reported that DNA enter the BPV E1 helicase in the direction of 3' to 5'^{12,13,2}. In our experiments, dsDNA with single strand arm at 3' was used as well. The description was revised as below:

Original: *It is reported that BPV E1 belongs to the SF3 class of helicases and has 3'-5' polarity when unwinding DNA.*

Revised: *It is reported that BPV E1 belongs to the SF3 class of helicases, and DNA enter the BPV E1 helicase in the direction of 3' to 5'.*

18) In Fig 2a dsDNA enters the wide end of the helicase. In Fig 3 cde, it enters the narrow end

Response: We are sorry for the mistake. The dsDNA entered the narrow end (N terminal) of the helicase nanopore, and Fig. 2a was modified accordingly in the revised manuscript.

19) It is unexpected that the helicase can insert into a bilayer in either orientation. Do the authors have an explanation.

Response: Since the helicase nanopore is not a native membrane channel, in combination of the structure analysis revealing a dispersed distribution of hydrophobic residues, we hypothesized that insertion could happen in either orientation. Both Cryo-EM characterization of helicase nanopore proteoliposomes vesicles [Figure R6] and single channel recordings of BLM experiments [Figure S14] confirmed this.

20) E1 306-577 has no unwinding activity--- does it transport ss or ds? --- the authors need to be clear which E1 they are talking about at all points in the paper.

Response: BPV E1 (306-577) without GST and BPV E1 (306-605) without GST mainly aggregated and denatured after cleavage in the process of purification, and their activity were tested by fluorescence quenching assay (Figure R9). It can be observed that both of them showed no helicase activity in contrast to BPV E1 (306-605) with GST. Thus BPV E1 (306-605) with GST was used as the helicase nanopore in the paper unless otherwise stated. [Text Revision]

Figure R9. Fluorescence quenching assay of different constructs of the BPV E1. a. BPV E1(306-605) with GST-tag; b. BPV E1(306-605) without GST-tag. c. BPV E1(306-577) with GST-tag.

21) What does linear dsDNA with sticky ends mean here?-- show the constructs used in the paper

Response: Constructs of dsDNA used in the study were displayed in Fig. R4, and the description was clarified as below:

Original: Using linear dsDNA with sticky ends, we investigated whether the engineered helicase nanopore in lipid bilayer membrane (BLM) retains helicase activity or not.

Revised: Using linear dsDNA (sequence E, structure shown in Figure S10) with partially complementary sequence possessing a single strand arm, we investigated whether the engineered helicase nanopore in lipid bilayer membrane (BLM) retains helicase activity or not (buffer condition: PBS, pH 7.0.).

22) The patterns seen for the interaction with dsDNA should be clearly described in the text and some attempt made to interpret them. The observations in reference 32 should also be clearly explained-- why should they be similar to what is seen in this paper? They do not seem to be relevant.

Response: Interpretation of the patterns was added in the revised manuscript as following:

. A schematic drawing (top right panel in Figure 3) was proposed according to the structure of the dsDNA and the function of the helicase nanopore, showing the three stages of dsDNA unwinding and translocation through the helicase nanopore corresponding to the three steps in the current recording events: 1. The capture of the 3' end of the single strand arm by the helicase nanopore, 2. the unwinding process of double strands, and 3. the release of single strand DNA. Additionally, the number of blockage steps can vary from 1 to 3.

In the reference¹⁴, translocation of ds–ss–ds DNA complexes and barcoded ssDNA through solid-state nanopores were studied. The latter showed two types of events: shallow and fast ssDNA translocation events, and multi-level and slow barcoded ssDNA translocation events. (Figure R10 left). The dsDNA with single strand arms used in our study were translocated through the helicase nanopore, and the three stages of current level are comparable to that of barcoded ssDNA in the reference. The difference is that the barcoded ssDNA (3-bp out of 22-nt) showed a much shorter dwell time of deep blockage than dsDNA in our study (24 complementary base pairs out of 44-nt).

Figure R10. Signal comparison between barcoded ssDNA translocation through solid-state nanopores¹⁴ (left) and dsDNA-helicase nanopore interaction (right).

23) The translocation rate of DNAs into cells should be quantified and compared with the event rate in planar bilayers.

Response: Following the suggestion, we made a comparison between the translocation frequency of helicase nanopore in cell membrane and in the planar bilayer under the same ssDNA concentration (200 nM) and voltage. The result (Fig. R11) shows a rate of $36.7 \pm 11.37/s$ (n=221 from 3 intendent trials with a total recording time of 6.0 s) for cell membrane experiments, and $25.6 \pm 6.68 /s$ (n=730 from 3 intendent trials with a total recording time of 28.5 s) for planar bilayer experiments, which are comparable.

Figure R11. The comparison between the ssDNA translocation events frequency of helicase nanopore in HEK293T cell membranes and in planer lipid bilayer. Typical current trace recording were presented: a. Cell membrane experiments (-60 mV bias voltage, 200 nM ssDNA in pipettes). b. Planar bilayer experiments (+60 mV bias voltage, 200 nM ssDNA). c. Histogram of translocation events rates (n=221 from 3 intendent trials with a total recording time of 6.0 s for cell membrane experiments, and n=730 from 3 intendent trials with a total recording time of 28.5 s for planar bilayer).

24) Top: 76.3 pS-- what 'channel' is this?

Response: The 'channel' referred to helicase nanopore in cell membrane. We have updated the description in the revised manuscript text to avoid duplication with the native cell membrane channels.

25) "the blockage signals represented the molecular transport extracted from inside the cell" What does this mean? What was transported?

Response: Along with the HEK293T control which shows no specific blockage events (top trace in Figure 4e), transient blockage signals were observed in the same experiment condition for a helicase nanopore embedded in the cell membrane (bottom trace in Figure 4e). Those control experiments revealed that the signals were associated with the cytoplasm in which water, ion and macromolecules are the major content in cytosol. In the setup of a positive charge applied to the electrode outside the cell membrane, negatively charged macromolecules was expected to transport towards the helicase nanopore from inside the cytoplasm, which could be proteins, messenger RNA, transfer RNA and ribosomal RNA. Analysis and statistics of the signals showed dwell time and blockage percentage comparable to that ssDNA (Panel d and f in figure 1), and this could be a indication of RNA transport. Investigations to those interesting questions could be conducted in future related studies.

26) "genomic DNA" -- what is intended here?

Response: The genomic DNA refers to all DNA fragments from the incubated HEK293T cells. Q-PCR experiments on those sample would allow the quantification of ssDNA delivered into the cell. 'Total DNA from the incubated HEK293 cell' was used instead in the revised manuscript.

27) ssDNAs transported into cells--- which DNA used?

Response: ssDNA Sequence F (5'- TGTTT TGCGAACTCC CCAAT ACTTT TCTTT TCAAA TTAAAATCT GCTCC TCACC CGCCT TTTTC TCTAA TGCCT GGAAG-3') was used. It was described in the methods section of 'Transfection, DNA extraction and Q-PCR', and name of the sequence (Sequence F) was added to the description in the revision.

28) Explain how helicase and unwinding activities differ

Response: There is no essential difference between helicase and unwinding for the helicase nanopore to separate the dsDNA into two complementary strands. When introduced to this work,

unwinding refers to the strands separation process by membrane-embedded helicase nanopore whereas helicase activity refers to the process *in vivo* and fluorescence quenching assay *in vitro*.

29) What do "transfection" and "genomic DNA" mean here?

Response: We used the term 'transfection' to describe the process of introducing ssDNA into HEK293T cells using the helicase nanopore.

'Genomic DNA' means the 'total DNA from the incubated HEK293 cell', and the description was updated as mentioned in response to Q26.

30) Fig 1d-- what is the evidence for the structure shown? Where are the hydrophobic residues on the helicase? Is the interaction demonstrated by Stoddart more likely [3].

Response: The helicase nanopore in lipid bilayer can be confirmed by the images of additional Cryo-EM characterization of helicase nanopore proteoliposomes vesicles (Figure 1e and R6), and analysis in Figure R7.

The hydrophobic residues were labeled with green in Figure R5 in Response to Q11.

As mentioned in Response to Q13, about half of the helicase nanopore interacted with lipid bilayer as surface attachment bilayer (similar to the truncated α -hemolysin pores), and another half were inserted into the bilayer with varying depths.

31) Fig 1ef- Put e and f on the same scale. Indicate the structures that are thought to be helicase molecules.

Response: We updated the figure scale following the suggestion, and the helicase molecules were highlighted with blue dashed outlines in insets of Figure R6.

32) Fig 2-- Why are b and c at different salt concentrations?

Response: To investigate the helicase nanopore systematically, we performed conductance measurements in various conditions, and current traces in panel b was an example of the result. Besides the condition of 1 M KCl in panel c, we also studied the conductance of the helicase nanopore at 0.5 M KCl, 0.3 M KCl and PBS. With increased salt concentration, the conductance increased in a way similar to other protein nanopores.

33) Fig 2 e--- Show an expanded segment so that the nature of individual blockades can be discerned

Response: Magnified view showing individual blockages is presented in revised Figure 2e as suggested.

Figure R12. (Fig. 2e). A typical current trace with blockage events caused by 100 nM ssDNA was added in 0.5 M KCl, 5 mM HEPES, pH 7.5 ($n > 3$).

34) Fig 3: show the zero current level on each trace

Response: Since both positive and negative bias voltages were used for helicase nanopore experiments in BLM and cell, we added baseline current level (close status) and opening level in all traces where applicable. Source data used to generate the traces can be found in the Source Data file provided online.

35) Step 3 (in b- e) seems longer than Step 1. This is not the case in the histograms

Response: As shown in Fig. R13, the mean dwell time of third step (7.13 ± 3.38 ms) was indeed longer than the first step (9.23 ± 7.57 ms).

Figure R13. The dwell time comparison of step 1 and step 3. Step 3 do has longer dwell time than step 1 in most of translocation events. Experiment condition: PBS, pH 7.0, 120 mV, 15 nM dsDNA (sequence E), $n = 79$.

36) Fig 3: The nature of the events underlying the current levels is proposed in the structures in the top right--- these ideas are not discussed in the text.

Response: Following discussion was added to the Results & Discussion section of the revised manuscript:

. A schematic drawing (top right panel in Figure 3) was proposed according to the structure of the dsDNA and the function of the helicase nanopore, showing the three stages of dsDNA unwinding and translocation through the helicase nanopore corresponding to the three steps in the current recording events: 1. The capture of the 3' end of the single strand arm by the helicase nanopore, 2. the unwinding process of double strands, and 3. the release of single strand DNA. Additionally, the number of blockage steps can vary from 1 to 3.

37) The origin of the blockades in Fig 4e is unclear

Please see response to Q25.

38) There is no abstract

Response: Nanoscale transport through nanopore and live-cell membrane plays a vital role in both key biological processes as well as biosensing, DNA sequencing and single-cell analysis *in-situ*. Active translocation of DNA through these nanopores usually need enzyme assistance. Here we present a new class of nanopore derived from truncated helicase E1 of Bovine papillomavirus (BPV) with a lumen diameter of c.a. 1.3 nm, named helicase nanopore. Engineering of the pore with GST tag improves its membrane compatibility, and Cryo-Electron Microscopy (Cryo-EM) imaging of their proteoliposome vesicles reveal the interaction of both surface attachment and membrane insertion. The helicase nanopore reconstituted in planar lipid bilayer could not only act as a conductive pore to allow the translocation of ssDNA, but also retain the ability to unwind double-stranded DNA (dsDNA) *in vitro*. The measured conductance of the helicase nanopore was c.a. 1.34 nS in 1 M KCl, pH ,7 and the mean unwinding speed for dsDNA was 65.54 ± 48.06 bp per second under +120 mV. The helicase nanopore activity could be inhibited by the removal of Mg^{2+} or adenosine triphosphate (ATP). Furthermore, we incorporate this helicase nanopore into the live cell membrane of HEK293T, and monitor the ssDNA delivery into the cell real-time at single molecule level. This new type of nanopore is expected to provide an interesting tool to study

the biophysics of biomotors *in vitro*, with potential applications in biosensing, drug delivery and real-time single cell analysis.

39) These are remarkable results and the editors should ensure that the authors make their constructs available so that the results can be duplicated by others.

Response: We appreciate the reviewer's expertise and constructive comment, and work at our best to address the questions. All the protein construct, DNA sequences, and detailed description of the experiment protocols are provided in the manuscript. The source data are provided as a Data Set online readily available for the researchers in the community.

Reviewer #3 (Remarks to the Author):

In their work, Sun et al report that the E1 helicase from Bovine Papilloma Virus can be inserted into lipid membranes and be used as a nanopore to translocate ssDNA. The authors performed a series of in vitro and in cell electrophysiological experiments to prove the activity of the inserted helicase. Remarkably, they show that under certain conditions ssDNA translocation can be performed in an active manner, fueled by ATP hydrolysis. This is a significant advance in the field, which can be of potential interest for the community. However, there are several important issues which should be addressed before considering the work for publication.

Response: We appreciate the review's constructive comments and suggestions, which are very helpful for us to improve the work and the manuscript.

Main points:

1) Helicase incorporation and engineering: Helicase incorporation into the membrane is not directed and non-specific, which impedes the authors to embed the helicase in the membrane in a preferred orientation and favors short residence times (time helicase bound to membrane). Taken together, these limitations hinder the possible applications of this system (see also comments about statistics below). The authors may comment.

Response: We performed additional characterization of the helicase nanopore interaction with lipid membrane using cryo-electron microscopy. From 67 incorporated helicase nanopores in 45 images, statistics revealed that 83.5% were N-terminus incorporation, and 16.5% were C-terminus incorporation. Similar with other commonly used protein nanopores, the helicase nanopore incorporation with lipid bilayer are non-specific. For specific binding with certain type of cells or certain location in a cell, protein labeling or engineering is possible. We would be happy to investigate this in future follow-up studies..

2) The authors should explain briefly why BPV E1 306-577 and 306-605 fragments were chosen and show how deletions affect the in vitro activity of the enzyme. This is not shown.

Response: Rationale of the protein construct was discussed in the the revised manuscript following the reviewer's suggestion as below:

Original: In this research, two BPV E1 truncation of 306-577 and 306-605 were reconstructed with N-terminal glutathione s-transferase (GST) fusion protein into pGEX-6P-1 plasmid and expressed in Escherichia coli (E. coli) respectively (Fig. S2a, b). Both the two expressed proteins

were purified and then identified by sodium dodecyl sulfate polyacrylamide gel electrophoresis (SDS-PAGE) (Fig. 1c). With previously research (15-JVI), a C-terminal 28-aa peptide is important for complex formation by E1₃₀₆₋₆₀₅.

Revised: In this study, two kinds of BPV E1 truncated protein containing 306-577 fragment (BPV E1 helicase domain) and 306-605 fragment (BPV E1 helicase domain and C-terminal tail) were reconstructed with N-terminal glutathione s-transferase (GST) fusion protein into pGEX-6P-1 plasmid and expressed in *Escherichia coli* (*E. coli*) respectively (Supplementary Table 1, Fig. 2 a, b). Both expressed proteins were purified by size exclusion chromatography (Supplementary Fig. 2 c, d) and then identified by sodium dodecyl sulfate polyacrylamide gel electrophoresis (SDS-PAGE) (Fig. 1c). Compared with E1₃₀₆₋₅₇₇ (with GST-tag) and E1₃₀₆₋₆₀₅ (with GST-tag cleaved) which tended to precipitate, E1₃₀₆₋₆₀₅ with GST-tag was more stable. Hence we chose E1₃₀₆₋₆₀₅ truncated protein with GST-tag as helicase nanopore in the following studies unless otherwise specified.

3) In order to increase the solubility of the protein, the authors fused a GST-tag to the above constructs. However, it is not stated which constructs were used in each of the experiments shown in the manuscript. The effect of the GST-tag in the *in vitro* helicase activity is also not reported.

Response: The GST-tag was added to the N-terminal of helicase nanopore, which improves its stability. As shown in Fig. 3a and Fig. S11, we had conducted the helicase activity of BPV E1 (306-577) and BPV E1 (306-605, with and without GST-tag). The result demonstrated that BPV E1 (306-577) and BPV E1 (306-605, without GST-tag) had little helicase activity. Then we choose BPV E1 (306-605, with GST-tag) as the helicase nanopore. We have added the relative description in the revised manuscript.

4) The putative role of lipid composition on helicase insertion into GUVs is not explored nor discussed.

Response: Synthetic lipid diphytanoyl phosphatidylcholine (DPhPC) was used in the experiments. Following the reviewer's suggestion, we investigated the helicase nanopore insertion using *E. coli* total lipid extract. Under similar experiment conditions, *E. coli* total lipid extract showed similar insertion efficiency and membrane stability compared with DPhPC. Discussion on this was added to the revised manuscript.

5) What is the average residence time of the nanopore in GUVs or ‘synthetic’ membranes?

Response: The mean residence time of the helicase nanopore in planar lipid bilayer was summarized in Figure R14 (S7), with a mean of 26.31 min.

Figure R14. The mean residence time of the helicase nanopore in planar lipid bilayer. Buffer condition: 0.5 M KCl, 5 mM HEPES, pH 7.5, with applied ranging from +150 mV to +200 mV (n=79).

6) Crio-EM images are not conclusive and hard to see. Same magnification should be used in control and no-control images.

Response: We carried out additional Cryo-EM characterization of the helicase nanopore proteoliposomes. High resolution images of 67 helicase nanopores from 45 images were collected (Please see Figure R6 and R7).

7) ssDNA translocation: Duplication of bias voltage (+50 to +100 mV) increased the frequency of ssDNA translocation events a merely 1.5%, from 81 to 82.5%. Of note is that at 100 mV the distribution (figure2h) is broader, implying higher error. So, it seems there is no significant correlation between the number of events and voltage, which is odd; however the authors stated this is a, why is that?

Response: The value 81% and 82.5% in panel f and h of Figure represented the blockage percent of the ssDNA translocation, which can be calculated as I_b / I_o (I_b : blocked current; I_o : open pore current). The wider distribution¹⁵ is caused by the more different blockages occurring under a high voltage.

We also analyzed the translocation events frequency, and positive correlation relationship with voltage can be observed (Figure R15).

Figure R15. (S9). The correlation between translocation events frequency and voltage. Experiment condition: 0.5 M KCl, 5 mM HEPES, pH 7.5, 250 pM ssDNA (Sequence A) (n≥3).

8) dsDNA unwinding experiments: The authors stated that 3 steps are observed during dsDNA unwinding by the helicase. However, at least 4 different steps are clearly distinguishable in each trace (an additional step is clearly visible after step 3). The authors may comment on the nature of these steps. How do the properties of each step depend on ATP concentration, temperature, DNA length, etc? What is the variability of these steps from one trace to another?

Response: a. We have replaced the signal diagram for more accurate description according to your suggestion (Fig. R16). The three blockage levels of the signal were caused by a. capture of single strand DNA arm by helicase nanopore, b. unwinding process, c. the release of ssDNA.

Figure R16. A typical current pattern induced by dsDNA translocation/unwinding (Buffer condition: PBS, pH 7.0, 120 mV, sequence E).

b. Following the suggestion, we analyzed the dependence of dwell of the 3 steps on ATP concentration, temperature, DNA length (Figure R17). It can be observed that Longer DNA (44 bp vs. 24 bp) caused a longer dwell time of all three levels, indicating a longer time needed for capture and unwinding. Lower temperature (12°C vs. 32 °C) slowed down all three levels. Interestingly, increased ATP concentration speeded up the unwinding step., but slowed downed the capture and exit step slightly.

Figure R17. The effect of DNA length on each step of unwinding signals (left, 24 bp DNA: sequence E, 44 bp DNA: sequence H, PBS, pH 7.0, 120 mV.)a: The effect of temperature on each step of unwinding signals (middle, sequence H, PBS, pH 7.0, 120 mV.). c: the effect of ATP concentration on each step of unwinding signals (right, sequence E, PBS, pH 7.0, 120 mV). (n =11 for each data point).

c. From panel c,d,e in Figure 3, and Figure R18, we can see the signal of dsDNA translocation through the helicase nanopore had a wider variation compared with ssDNA translocation.

Figure R18. The diversity among different single unwinding signals was mainly on different dwell time of blockage levels, especially on the biggest blockage levels (PBS, pH 7.0, 120 mV, sequence E).

9) Overall, the work will increase in scope if the authors prove that their method is robust enough for detailed biophysical characterization of the helicase activity.

The authors reported an average dsDNA unwinding rate of 64.54 +/- 48.06 bp/s. So, the average error seems to be ~75%. However, the data presented in figure 3h (duration time vs. length of dsDNA) presents very small error bars, suggesting very homogeneous unwinding rates between different nanopores. How is this possible? Also, data in figure 3h indicate an average unwinding

velocity of ~13-16 bp/s, which is ~4 times slower than the reported average dsDNA unwinding rate (64.54bp/s). On the other hand, how do these rates compare with unwinding rates *in vitro*?

Response: The velocity range (65.54 +/- 48.06 bp/s) in the text refer to the unwinding velocity of sequence E (24 bp) under 120 mV (the data point in the red box in Figure 3h, shown in Figure R19.), the unwinding velocity of sequence H (44 bp) under 120 mV is range from 17 bp/s to 23 bp/s and the unwinding velocity of sequence B (80 bp) under 120 mV is range from 12 bp/s to 13 bp/s. Under the same condition, longer sequence has slower unwinding velocity.

Figure R19. The data point in Figure 3h and the dsDNA unwinding velocity distribution under this condition.

Table R3. Comparison of dsDNA unwinding rates by different helicases *in vitro*.

Helicase	Unwinding velocity / bp·s ⁻¹	Condition
PcrA ¹	38 ± 24	-
RecBCD ¹	200 ± 175	25°C
AddA ^N B ^N ¹	82 ± 63	-
T7 (M64L) ²	15	2 mM dTTP, 18°C
DmCMG ³	0.1 ± 0.08 to 0.47 ± 0.56	0.05 mM ATP to 4 mM ATP

10) Importantly, the statistics (number of unwinding events) in each experimental condition are not indicated. The authors should indicate clearly the statistic in all conditions.

Response: We have clearly described all the experimental conditions in the revision.

11) Error bars are not defined (both for figure 3 and figure 4). Do they show s.d., s.e.m.?

Response: In this work all errors bars are from standard deviation (s.d.). Relative descriptions were added in the revised manuscript.

12) There is some confusion about dwell times (figure 3f), duration times (figure 3h) and average velocities. The authors should use uniform terminology and clarify these points. Figure 3f shows dwell times vs voltage, but for what DNA length? Figure 3h shows duration time vs length of dsDNA, but for which voltage?

Response: We change them to dwell time in the revision. For figure caption, 24-bp DNA length (sequence E) in Fig. 3f and 120 mV of bias voltage in Fig. 3h were added.

13) The temperatures reported in the main text are different from the temperatures indicated in the plots (figure 3i and 3j). According to the main text, Figures 3i and 3j should correspond to figures 3k and 3l (and the other way around).

Response: We are sorry for the mistake, and have corrected this in the revised manuscript.

14) The authors should comment on and elucidate all these points carefully. As mentioned, proper/rigorous biophysical characterization of the nanopore is key to validating their method. Overall, the presentation of the biophysical characterization of the nanopore is very confusing.

Response: We improve the presentation with additional data, analysis and accurate description with necessary information included as suggested.

15) Incorporation and activity measurements of helicase in cell membranes:

The reported efficiency of the method to internalize ssDNA into cells using the helicase-nanopore is ~5% (5 out of 98 cells internalized ssDNA upon membrane modification with the nanopore) and the residence time of the helicase at the cell membrane is short (not specified). These are important drawbacks of the method that will limit its possible applications in the future. The authors may comment.

Response: We agree with the comment on the efficiency of ssDNA internalization into cell, and short residence time. This is mainly due to the its low membrane affinity compared with other native membrane channels. In order to improve this, further protein engineering is necessary. On the other hands, short residence time on membrane would significantly reduce its toxicity to the

cell. Depending on the purpose of application, for example drug delivery, single cell analysis, target therapy to tumor cells, specific interaction could be designed. Those discussion and comments were added in revised manuscript.

16) The number of control experiments seems not sufficient. Only 8 cells were used as controls. Considering that the efficiency of the method is ~5%, controls should include more than 8 cells.

Response: We have added more control groups. As shown in Fig. R20 for those control groups, the concentration of ssDNA was 200 nM, and no helicase nanopore was added. The signals were recorded under -60 mV. No obvious insertion or translocation signals were observed in those 20 traces.

Figure R20. The current trace of the control group on HEK293T cell membrane. (Intracellular solution: 150 mM KCl, 1 mM EDTA, 10 mM HEPES, pH 7.4. Extracellular solution: 145 mM NaCl, 10 mM HEPES, 10 mM glucose, 4 mM KCl, 2 mM CaCl₂, 1 mM MgCl₂, pH 7.4. Voltage was -60 mV. The concentration of ssDNA in pipettes was 200 nM. No helicase nanopore was added in pipettes).

17) What is the actual residence time of the nanopore in cell membranes? The authors stated in the main text that it is less than 10 seconds, the figures show time windows < 2 seconds.

Response: The modified figure about the residence time of the nanopore in cell membranes had been shown below. Although majority of the helicase nanopore just stay on cell membrane for a short time (less than 2s), but there were still insertions stable for much longer time (Fig R21 and Fig S15).

Figure R21. (F4b). The histogram of duration time when the helicase nanopore inserted in a HEK293T cell membrane.

18) The authors suggested the possibility of transport of intracellular material outside the cell by the nanopore when a positive voltage is applied (page 6, line 172). How do the authors know that the observed blockage events correspond to transport events? Could it just be that the pore gets clogged with intracellular material migrating to membrane due to voltage application?

Response: There could be blockage events caused by the pore getting clogged with intracellular material, especially the blockage events with long blocked time. But majority of the signals have a similar blockage percentage and dwell time with the translocation events in planar bilayer experiments. This indicated that intracellular material transport outside the cell through the helicase nanopore is possible.

19) The procedure to determine ssDNA translocation into cells using ATP (figure 4h) and its validation are hardly described.

Response: Description about the procedure and validation of determining ssDNA translocation into cells using ATP (Fig. 4h) was updated in in the revised manuscript as following:

Original: The transport of ssDNA into HEK293T cells via the helicase nanopore was observed when ATP was present. However, without ATP the translocation did not occur (Fig. 4h). Therefore, it was concluded that ATP was needed for the helicase nanopore translocation of ssDNA through live cell membranes.

Revised: For control groups of DMEM, ALB and helicase nanopore only, no expression of ssDNA was observed. When incubated with ssDNA and ATP, helicase nanopore showed higher expression level of ssDNA compared with DMEM and ALB. (Fig. 4h). In the case of incubation with ssDNA and without ATP, the expression of ssDNA was much lower and showed no significant difference DMEM, ALB and the helicase nanopore (lower panel of Fig. 4h). Those results proved that the ATP was necessary for ssDNA translocation into cells by the helicase nanopore.

20) Overall, in its current state, the work seems a collection of experiments; accurate, meticulous characterization of the nanopore in each condition is missing. The manuscript will benefit greatly from English editing, reorganization and a more detailed explanation of experimental conditions in each case.

Response: With the suggestions and comments from the reviewer, we made substantial revision with extensive data, in-depth analysis and discussion to present the work accurately. Thank you!

REFERENCES

1. Kurg, R. *et al.* Bovine papillomavirus type 1 E2 protein heterodimer is functional in papillomavirus DNA replication in vivo. *Virology* **386**, 353–359 (2009).
2. Chaban, Y. *et al.* Structural basis for DNA strand separation by a hexameric replicative helicase. *Nucleic Acids Res.* **43**, 8551–8563 (2015).
3. Stoddart, D., Heron, A. J., Mikhailova, E., Maglia, G. & Bayley, H. Single-nucleotide discrimination in immobilized DNA oligonucleotides with a biological nanopore. *Proc. Natl. Acad. Sci.* **106**, 7702–7707 (2009).
4. Derrington, I. M. *et al.* Subangstrom single-molecule measurements of motor proteins using a nanopore. *Nat. Biotechnol.* **33**, 1073–1075 (2015).
5. Wang, Y. *et al.* Nanopore Sequencing Accurately Identifies the Mutagenic DNA Lesion O 6 - Carboxymethyl Guanine and Reveals Its Behavior in Replication. *Angew. Chemie Int. Ed.* **58**, 8432–8436 (2019).
6. Harrington, L., Alexander, L. T., Knapp, S. & Bayley, H. Pim Kinase Inhibitors Evaluated with a Single-Molecule Engineered Nanopore Sensor. *Angew. Chemie Int. Ed.* **54**, 8154–8159 (2015).
7. Stoddart, D. *et al.* Functional truncated membrane pores. *Proc. Natl. Acad. Sci.* **111**, 2425–2430 (2014).
8. Hall, J. E. Access resistance of a small circular pore. *J. Gen. Physiol.* **66**, 531–532 (1975).
9. Sanders, C. M. *et al.* Papillomavirus E1 helicase assembly maintains an asymmetric state in the absence of DNA and nucleotide cofactors. *Nucleic Acids Res.* **35**, 6451–6457 (2007).
10. De-Donatis, G. *et al.* Finding of widespread viral and bacterial revolution dsDNA translocation motors distinct from rotation motors by channel chirality and size. *Cell Biosci.* **4**, 30 (2014).
11. Enemark, E. J. & Joshua-Tor, L. Mechanism of DNA translocation in a replicative hexameric helicase. *Nature* **442**, 270–275 (2006).
12. Seo, Y. S., Muller, F., Lusky, M. & Hurwitz, J. Bovine papilloma virus (BPV)-encoded E1 protein contains multiple activities required for BPV DNA replication. *Proc. Natl. Acad. Sci.* **90**, 702–706 (1993).
13. Lee, S. *et al.* Dynamic look at DNA unwinding by a replicative helicase. *Proc. Natl. Acad. Sci.* **111**, E827–E835 (2014).
14. Liu, K. *et al.* Detecting topological variations of DNA at single-molecule level. *Nat. Commun.* **10**, 3 (2019).
15. Khulbe, P. K., Mansuripur, M. & Gruener, R. DNA translocation through α -hemolysin nanopores with potential application to macromolecular data storage. *J. Appl. Phys.* **97**, 104317 (2005).

Reviewers' Comments:

Reviewer #1:

Remarks to the Author:

The authors did a good job with the revised version, and the manuscript is ready for publication in my opinion. Question: what is alpha-hederin?

Reviewer #2:

Remarks to the Author:

The authors have made a strong effort to improve their presentation. While they have succeeded to an extent, there are still several loose ends, but I think this is to be expected in a paper that presents genuinely new findings, which is the case here.

In reviewing the revised paper, the SI and the Reviewers comments and the replies, I have come up with a few bits and pieces, but I do think the results should be published at this point after any obvious issues have been remedied. The writing could do with a final polishing too.

Reviewer 1 and DNA capture. The authors might take a look at [1].

Abstract: "... the GST tag improves membrane compatibility ..." I don't think so-- it appears to improve solubility.

Abstract and throughout: Check the significant figures. I don't think the unwinding speed was determined to 4 sf. And, for example, on page 5-- what does 81.85 ± 0.00077 mean?

Ref 22 is interesting but this citation is misleading. Nanopore sequencing including the identification of modified bases was enabled by Oxford Nanopore Technologies (<https://nanoporetech.com>).

The "insertion" of the truncated helicase into the bilayer remains very puzzling. However, I do feel that the authors have now presented the data clearly and further work will be needed to clarify the situation. They might further mark the N and C termini on Fig S1 and the hydrophobic residues on Fig 1.

page 4 (Helicase nanopore in lipid bilayer...). Give the concentration of the helicase used.

page 5: alpha-Hederin-- Introduce alpha-Hederin or give a reference

page 5: "efficiency of insertion". Define the meaning of this phrase in the text.

page 5: "sequence A". Give the sequence or at least the length of the DNA in the text.

page 5, Fig S6: Give the conditions. In particular, the helicase concentration.

page 5, 6: "burst"- poor usage. In electrophysiology, bursts are short series of events that separated by quiet intervals. I don't think the authors mean this.

page 5: "non-specific spikes" What does this mean here?

page 6 and throughout: If the pores can be oriented in either direction, the DNA entry will depend on the side of addition. Thus some pores will appear to be inactive. This issue should be clarified.

page 8, Fig 4: Figs 4c and e imply that an inside-out patch was pulled off the cell. I don't think this is the case. The situation should be clarified.

page 9: "This confirmed the real-time through the helicase nanopore" The authors might note that the applied potential is opposite to the resting potential of the cell, so that "delivery" could not be done without the pipet.

page 9: The nature of the transport from inside the cell is not clarified in the text.

page 10: The helicase transport into the cell is certainly interesting. The authors state that they use Q-PCR, so they should be able to calculate the mean number of DNAs translocated per cell. Also sequence B is a ssDNA, which is surprising. "Expression level" here is also confusing.

Fig 2c: give the applied potential

Fig 2e: state which DNA was used

1. Maglia G, Rincon Restrepo M, Mikhailova E, Bayley H: Enhanced translocation of single DNA molecules through α -hemolysin nanopores by manipulation of internal charge. Proc Natl Acad Sci U S A 2008, 105:19720-19725.

Reviewer #3:

Remarks to the Author:

The authors have successfully addressed all the issues previously raised. The data presented in the revised manuscript are convincing and overall, the work could influence the thinking in the field. The in vitro characterization of the helicase activity of the different nanopore constructs used in this work would have been interesting (to compare it with the reported activity upon membrane incorporation). I hope the authors work to address this question in future works.

Response to the Reviewers comments

We would like to thank the reviewers for their time and expertise. Their comments and suggestions are very helpful for us to finalize the manuscript. A point-by-point response to the comments is listed below.

Reviewer #1

1) The authors did a good job with the revised version, and the manuscript is ready for publication in my opinion. Question: what is alpha-hederin?

Response: We are happy that our efforts are recognized by the reviewer.

Alpha-hederin (Ah) is a natural pore-forming amphiphilic compound from the saponin family, and was reported by Jeong et al. recently for its application as a nanopore sensor. The following citation was added to the manuscript as Reference #38.

[Ref#38] Jeong, K.-B. et al. Alpha-Hederin nanopore for single nucleotide discrimination. *ACS Nano* **13**, 1719-1727 (2019).

Reviewer #2

The authors have made a strong effort to improve their presentation. While they have succeeded to an extent, there are still several loose ends, but I think this is to be expected in a paper that presents genuinely new findings, which is the case here.

In reviewing the revised paper, the SI and the Reviewers comments and the replies, I have come up with a few bits and pieces, but I do think the results should be published at this point after any obvious issues have been remedied. The writing could do with a final polishing too.

Response: We appreciate the reviewer's expertise and remarkable insights. The manuscript was carefully revised to address those remaining questions. The language and text were double-checked to ensure accuracy and clarity.

1) Reviewer 1 and DNA capture. The authors might take a look at [1].

Response: Following the suggestion, we discussed the DNA translocation frequency in the manuscript with the reference as below:

The frequency of DNA translocation through protein nanopore could be increased by augmenting the internal positive charge of the nanopore⁴¹, and it was interesting to found that the frequency of ssDNA translocation events through the helicase nanopore was higher compared with alpha-hemolysin even with internal positive charge augment under similar condition⁴¹. The reason for faster DNA capture rates could be attributed to the nature of the virus-derived helicase motor to process and unwind dsDNA in host cell during infection, in contrast to the bacteria derived pore-forming toxin nanopores.

[Ref#41] Maglia, G., Restrepo, M. R., Mikhailova, E. & Bayley, H. Enhanced translocation of single DNA molecules through α -hemolysin nanopores by manipulation of internal charge. *Proc. Natl. Acad. Sci.* **105**, 19720-19725 (2008).

2) Abstract: "... the GST tag improves membrane compatibility ..." I don't think so-- it appears to improve solubility.

Response: We agree with the comment. The GST-tag indeed improve the solubility and stability of the helicase nanopore, thus leading to a stable membrane fusion. The relevant description was revised for accuracy in the manuscript text.

3) Abstract and throughout: Check the significant figures. I don't think the unwinding speed was determined to 4 sf. And, for example, on page 5-- what does 81.85 ± 0.00077 mean?

Response: a. We thank the reviewer for pointing this out. The unwinding speed by the helicase nanopore in vitro showed relatively large variation and was calculated to be 65.54 ± 48.06 bp/s (obtained from the equation $v_{\text{unwinding}} = n_{\text{number of base pairs}}/t_{\text{dwell time}}$), which showed one significant figure. Thus, we corrected the description in the revision:

The calculated unwinding speed for dsDNA was 7 ± 5 bp per 100 ms under +120 mV.

b. 81.85 ± 0.00077 % was obtained from the peak value of Gaussian fitting of current blockage events by ssDNA (100 nM, sequence A of 48 nt) translocating through the helicase nanopore, which showed five significant figures. The description was updated as below:

Gaussian fitting of these current blockage events gave the peak value of (I_b/I_o) 81.850 ± 0.001 %.

There are also previous nanopore studies with similar statistical method and results, for example: The histogram of the relative blockade current I_b/I_o values induced by the mixture of arginine peptides of 6 different lengths is shown in Fig. 1d. Well-separated peaks corresponding to distinct populations

with preferred I_p/I_o values are observed. For each of the 6 main populations, the mean relative blockade current is (mean value of a gaussian fit of the peak \pm one standard deviation): 0.234 \pm 0.001, 0.286 \pm 0.002, 0.353 \pm 0.002, 0.435 \pm 0.002, 0.530 \pm 0.002, 0.631 \pm 0.004.

[Ref. R1] Piguet, F. et al. Identification of single amino acid differences in uniformly charged homopolymeric peptides with aerolysin nanopore. *Nat. Commun.* 9:966 (2018).

We carefully checked throughout the manuscript and supplementary information, and corrected similar mistakes related with significant figures.

4) Ref 22 is interesting but this citation is misleading. Nanopore sequencing including the identification of modified bases was enabled by Oxford Nanopore Technologies (<https://nanoporetech.com>)

Response: The following reference reporting the advanced nanopore sequencing technology and its emerging impact on real-time genomic surveillance was mentioned in the revision:

[Ref #22] Quick, J. et al. Real-time, portable genome sequencing for Ebola surveillance. *Nature* **530**, 228-232 (2016).

5) The "insertion" of the truncated helicase into the bilayer remains very puzzling. However, I do feel that the authors have now presented the data clearly and further work will be needed to clarify the situation. They might further mark the N and C termini on Fig S1 and the hydrophobic residues on Fig 1.

Response: Panel **a** of Supplementary Figure 1 and panel **d** of Figure 1 were updated as suggested.

6) page 4 (Helicase nanopore in lipid bilayer...). Give the concentration of the helicase used.

Response: The concentration of the helicase nanopore used in this section was provided as below:

Solution containing the helicase nanopore (final concentration of 2-10 ng/mL) was added to the buffer solution in the cis-side of the BLM chambers...

7) page 5: alpha-Hederin-- Introduce alpha-Hederin or give a reference

Response: The following citation about alpha-Hederin was added to the manuscript as Reference #38.

[Ref#38] Jeong, K.-B. et al. Alpha-Hederin nanopore for single nucleotide discrimination. *ACS Nano* **13**, 1719-1727 (2019).

8) page 5: "efficiency of insertion". Define the meaning of this phrase in the text.

Response: The insertion efficiency was defined as the ratio of number of trials with helicase nanopore inserted into lipid bilayer to the total number of trials in the revision.

9) page 5: "sequence A". Give the sequence or at least the length of the DNA in the text.

Response: The detailed sequence information was provided in Supplementary Table 2. Following the suggestion, we have added the length of all the DNA sequences used in the revised manuscript text.

10) page 5, Fig S6: Give the conditions. In particular, the helicase concentration.

Response: The information was added to the manuscript text and figure legend as suggested.

11) page 5, 6: "burst"- poor usage. In electrophysiology, bursts are short series of events that separated by quiet intervals. I don't think the authors mean this.

Response: "Large number" was used to replace "burst" in the revision as suggested.

12) page 5: "non-specific spikes" What does this mean here?

Response: Non-specific spikes mean occasional background noise or ssDNA random collision with the helicase nanopore rather than specific transport signals. This description was added to the manuscript text in the revision as below:

Analysis of the current blockage events indicated that the ssDNA caused both distinct blockage events, and non-specific spikes due to occasional background noise or ssDNA random collision with the helicase nanopore.

13) page 6 and throughout: If the pores can be oriented in either direction, the DNA entry will depend on the side of addition. Thus some pores will appear to be inactive. This issue should be clarified.

Response: Following the review's suggestion, description was added to the manuscript as below:

Some pores would appear to be inactive while dsDNA was added to one side of chamber.

14) page 8, Fig 4: Figs 4c and e imply that an inside-out patch was pulled off the cell. I don't think this is the case. The situation should be clarified.

Response: All the cell patch experiments were conducted in cell-attach mode in the study. The schemes in Figure 4c and 4e were updated to avoid confusion.

15) page 9: "This confirmed the real-time ... through the helicase nanopore" The authors might note that the applied potential is opposite to the resting potential of the cell, so that "delivery" could not be done without the pipet.

Response: We thank the reviewer for pointing this out. The description was updated as below:
This confirmed the pipette-assisted real-time, single-molecule voltage-driven delivery of ssDNA into an individual live cell through the helicase nanopore.

16) page 9: The nature of the transport from inside the cell is not clarified in the text.

Response: The transport from inside the cell was discussed in the previous revision of response to reviewers. We add the discussion in this revision:

Along with the HEK293T cells control which showed no specific blockage events (top trace in Figure 4e), transient blockage signals were observed in the same experiment condition for a helicase nanopore embedded in the cell membrane (bottom trace in Figure 4e). Those control experiments revealed that the signals were associated with the cytoplasm in which water, ion and macromolecules were the major content in cytosol. In the setup of a positive charge applied to the electrode outside the cell membrane, negatively charged macromolecules was expected to transport towards the helicase nanopore from inside the cytoplasm, which could be peptides, RNA or other small biomolecules, etc. Analysis and statistics of the signals showed a dwell time and blockage percentage comparable to that of ssDNA (Panel d and f in Figure 4).

17) page 10: The helicase transport into the cell is certainly interesting. The authors state that they use Q-PCR, so they should be able to calculate the mean number of DNAs translocated per cell. Also sequence B is a ssDNA, which is surprising. "Expression level" here is also confusing.

Response: a. We agree with the reviewer that the cell transport study is worthy of detailed investigation. The experiment of helicase transport into the cell was initially designed to test the

feasibility of helicase nanopore as a DNA carrier to assist ssDNA transport into cells, so relative quantification Q-PCR method was chosen. Since the readouts from Q-PCR were in the form of relative quantification, calculation of the absolute number of DNAs translocated per cell was not feasible in the given experiment setup. We would like to investigate this in future studies.

b. ssDNA of sequence B was used both in helicase transport into cell, and patch clamp experiments on single cell for side-by-side comparison. Future studies on the dynamics and kinetics transport of dsDNA into live cell membrane could be carried out.

c. "Amplification" was used to replace "Expression" to avoid confusion in the revised manuscript.

18) Fig 2c: give the applied potential

Response: We have added the applied potential in the caption of figure 2c and in the corresponding description in the revised manuscript.

19) Fig 2e: state which DNA was used.

Response: Sequence A of 48 nt in Figure 2e was described in the revision as suggested.

1. Maglia G, Rincon Restrepo M, Mikhailova E, Bayley H: Enhanced translocation of single DNA molecules through α -hemolysin nanopores by manipulation of internal charge. Proc Natl Acad Sci U S A 2008, 105:19720-19725.

Reviewer #3

The authors have successfully addressed all the issues previously raised. The data presented in the revised manuscript are convincing and overall, the work could influence the thinking in the field.

Response: The review's constructive comments are greatly appreciated.

The in vitro characterization of the helicase activity of the different nanopore constructs used in this work would have been interesting (to compare it with the reported activity upon membrane incorporation). I hope the authors work to address this question in future works.

Response: We totally agreed with this, and follow-up characterization of the helicase kinetics and dynamics in vitro is being planned and conducted using various single molecule techniques besides electrophysiological measurement. We would also like to share those materials and results with larger research community. Thank you again.